# The association between polypharmacy and health-related quality of life among non-dialysis chronic kidney disease patients

**Leonie Adjeroh**[1]*, **Todd Brothers**[2], **Khaled Shawwa**[3], **Mohammad Ikram**[4], **Mohammad A. Al-Mamun**[1]

**1** Department of Pharmaceutical Systems and Policy, West Virginia University, Morgantown, West Virginia, United States of America, **2** College of Pharmacy, University of Rhode Island, Kingston, Rhode Island, United States of America, **3** Department of Medicine, Section of Nephrology, West Virginia University, Morgantown, West Virginia, United States of America, **4** Department of Surgery, Penn State, Hershey, Pennsylvania, United States of America

* cadjeroh@mix.wvu.edu

## Abstract

**Data Availability Statement:** All relevant data are within the paper and its Supporting Information files.

### Background and objective

The United States government spends over $85 billion annually on treating non-dialysis chronic kidney disease (CKD). Patients with CKD are prescribed a multitude of medications to manage numerous comorbidities associated with CKD. Thus, this study aims to investigate the association between polypharmacy and health-related quality of life (HRQoL) in non-dialysis CKD patients.

### Methods

This cross-sectional study utilized data from the Medical Expenditure Panel Survey (MEPS) from 2010 through 2019. We classified polypharmacy into three groups based on the number of medication classes: ≤ 4 (minor polypharmacy), 5 through 9 (major polypharmacy), and ≥ 10 (hyperpolypharmacy). To measure HRQoL, a Physical Component Summary (PCS) and a Mental Component Summary (MCS) were obtained from the 12-item Short-Form Health Survey version 2 and Veteran's Rand 12 item. We applied multivariable ordinary least squares regression to assess the association between polypharmacy and HRQoL in non-dialysis CKD patients.

### Results

A total of 649 CKD patients (weighted n = 667,989) were included. Patients with minor polypharmacy, major polypharmacy, and hyperpolypharmacy were 22.27%, 48.24%, and 29.48%, respectively. Major polypharmacy and hyperpolypharmacy were significantly and negatively associated with lower PCS scores when compared with minor polypharmacy [Beta = -3.12 (95% CI: -3.62, -2.62), p-value<0.001; Beta = -4.13 (95CI: -4.74, -3.52), p-value<0.001]. Similarly, major polypharmacy and hyperpolypharmacy were significantly and negatively associated with lower MCS scores when compared to minor polypharmacy [Beta

**Funding:** The author(s) received no specific funding for this work.

**Competing interests:** The authors have declared that no competing interests exist.

= -0.38 (95% CI: -0.55, -0.20), p-value<0.001; Beta = -1.70 (95% CI: -2.01, -1.40), p-value<0.001]. The top 5 classes of medications used by CKD patients were antihyperlipi-demic (56.31%), beta-adrenergic blockers (49.71%), antidiabetics (42.14%), analgesics (42.17%), and diuretics (39.65%).

## Conclusion

Our study found that both major polypharmacy and hyperpolypharmacy were associated with lower HRQoL among non-dialysis CKD patients. This study highlights the need for further evaluation of the combination of medications taken by non-dialysis CKD patients to minimize unnecessary and inappropriate medication use.

## Introduction

Chronic kidney disease (CKD) is defined as an impairment of the kidney function or structure that is present for more than three months [1]. Most cases of CKD are irreversible, and treatments are used to stabilize or slow the progression to kidney failure. In the United States (US), the incidence of CKD continues to rise. In 2010, it was estimated that 20 million Americans had CKD [2], and by 2023, that number soared to 35 million, which accounts for approximately 14% of the US adult population [3]. CKD cases have steadily increased due to an aging population, leading to higher costs to the healthcare system. According to recent data, Medicare fee-for-service spending in non-End Stage Renal Disease (ESRD) surpassed $85 billion in 2019, representing 23% of all Medicare fee-for-service spending [4]. CKD remains a major global and national public health problem due to its high burden on morbidity and mortality and progression to ESRD. The two most common risk factors associated with CKD are diabetes and hypertension [5, 6], with the prevalence of CKD among individuals with diabetes and hypertension being 28% and 21%, respectively [7].

Multimorbidity is characterized by the presence of two or more diseases at the same time. It is common in patients with CKD specifically, conditions such as heart failure, anemia, ischemic heart disease, thyroid disorders, diabetes, and hypertension often occur concurrently with CKD [8]. The prevalence of multimorbidity increases with age [9, 10], leading to the concomitant use of multiple medications, known as polypharmacy [11]. Polypharmacy is increasingly becoming a potential problem for individuals with CKD, as they are often treated for other chronic comorbidities or multimorbidity. In a recent Japanese study, polypharmacy was associated with an increased risk of chronic conditions and mortality in older patients with CKD not undergoing dialysis [12]. The use of multiple medications is associated with adverse health outcomes among older adults with CKD [13], and patients with advanced stages of CKD have a higher burden of medication use than those without CKD [14]. The burden of polypharmacy can lead to inappropriate use of medications, which can harm patients with renal impairment [15, 16] and subsequently compromise their overall quality of life [17].

Health-related quality of life (HRQoL) is a vital assessment measure in patients with chronic diseases such as cancer, heart disease, and CKD. When determining Physical Component Summary (PCS) scores, a component of HRQoL, it was found that PCS scores were associated with polypharmacy among cancer survivors in the US [18]. Additionally, lower PCS scores were associated with polypharmacy among US adults with cardiometabolic risk factors [19]. Among hemodialysis patients, HRQoL was significantly associated with an increased mortality risk and hospitalization [20–22]. The Mental Component Summary (MCS) score is

another component of HRQoL. ESRD patients had significantly lower PCS and MCS scores compared to healthy participants (8.7 ± 0.8 points lower, p-value < 0.001 and 2.7 ± 0.8 points lower, p-value < 0.001) [23]. Patients with CKD have a significant impairment in HRQoL with a decline in PCS score for each progressive stages of CKD [24]. Multiple studies have shown that females and older patients (> 65) with CKD have lower PCS scores compared to males and younger patients [24, 25]. Among patients with ESRD, HRQoL has been associated with low income, low educational level, and unemployment [25]. The impact of HRQoL on ESRD patients, who mostly require dialysis has been well documented [20–25]. In contrast, only a few studies have evaluated the association of HRQoL and non-dialysis CKD patients. To date, only one study on dialysis patients from a Dutch dialysis center has investigated the impact of polypharmacy on the HRQoL in CKD patients. Dialysis patients had lower PCS scores when the medication threshold was greater than or equal to five and lower MCS scores when the number of medications was between 14–27 [26]. More studies are needed in this area to further elucidate this association. To the best of our knowledge, no studies have investigated the impact of polypharmacy on the HRQoL in non-dialysis CKD patients.

This study aims to determine the association between polypharmacy and HRQol among non-dialysis CKD patients. We hypothesize that higher polypharmacy will be associated with lower HRQoL. In addition, we aim to identify the top 5 prescribed medication classes consumed by non-dialysis CKD patients and examine their prevalence across age groups, sex, levels of polypharmacy, and HRQoL categories.

## Data & methods

### Research design

This was a cross-sectional study conducted using medical expenditure panel survey (MEPS) data from 2010–2019 to determine the association between polypharmacy and HRQoL among non-dialysis CKD patients.

### Data source

MEPS is a national representative survey data of the civilian non-institutionalized U.S. population that is sponsored by the Agency for Healthcare Research and Quality (AHRQ) [27]. MEPS collects data on healthcare use, expenditure, insurance coverage, prescription drug use, and HRQoL from individuals, families, and selected population subgroups. MEPS household component (HC) data is a sub-selection of individuals who participated in the previous year's National Health Interview Survey (NHIS). The MEPS data is based on a complex sampling design that uses clustering, stratification, and multistage disproportionate sampling to account for the disparity in smaller subpopulations. MEPS consolidates its dataset in panels, each consisting of 5 rounds spanning over two years. Interviews are conducted every six months through 30 months, and data is released each year with a nationally representative estimate for different attributes. Responses to questions are collected through a self-response interview using computer-assisted personal interviewing technology or self-administered questionnaires (SAQ) for participants 18 years and above. To prevent underestimation and bias from the data collection process, AHRQ assigns person-weighted values and variance estimation strata to each yearly data to account for nonresponse and national estimates for population totals and characteristics of individuals and their families [27]. MEPS HC data file is released as a full-year consolidated data file with associated weights and estimates or as a longitudinal file with weights and estimates for the combined two-years data collection process. This study utilized the yearly consolidated data files and the accompanying SAQ weights and variance estimation strata. The 10-year data was included to increase the sample size of the CKD population.

CKD and other comorbidities are recorded as current medical conditions if the respondent from an eligible household reported being diagnosed with a disease based upon a medical event such as a hospital stay, emergency room visit, prescribed medication, or health care provider visit. Information identifying medical conditions was written as text and transcribed by professional medical coders following the format of the international classification of diseases, ninth and tenth revision, and clinical modification (ICD-9-CM and ICD-10-CM) codes. Clinical classification codes are utilized with ICD codes to group data into meaningful categories. Responses on health conditions are converted into ICD-9-CM codes for data collected before and through 2015 and ICD-10-CM codes for data collected after 2015. We extracted demographic variables, socioeconomic status, HRQoL, and mental health conditions from the HC files for each study year. We obtained information on prescribed medications from the prescription medicine files and data regarding medical conditions reported by respondents from the medical condition files. We consolidated individual yearly data variable files into one file using unique identifiers.

## Study population

The study population included all non-dialysis CKD patients 18 years and older, living in the US and alive during the data collection period. We selected individuals with CKD using ICD-10 codes N18 (chronic kidney disease) and N19 (unspecified kidney failure) for datasets from 2016 through 2019 and ICD-9 codes 585 (chronic renal failure) and 586 (unspecified renal failure) for datasets from 2010 through 2015.

## Exclusion criteria

As MEPS data is a sub-selection of individuals who participated in the previous year's NHIS, MEPS collects data from every individual in the target area. It is possible for the status of some of these individuals to have changed (for example, to an institutionalized population since participating in the NHIS). If a change in status occurs, MEPS assigns negative weights, PCS, and MCS scores. Hence, we excluded all participants with a negative weight on the self-administered questionnaire (SAQ). The SAQ is a paper-and-pencil questionnaire that asks questions about health status. We also excluded patients deemed ineligible to participate in the SAQ questionnaire based on the criteria from MEPS or if they had a negative PCS or MSC score. We excluded patients if they were < 18 years old, institutionalized, or had no information available for the specific round. Furthermore, we excluded patients if they had any cancer, were undergoing dialysis (identified from the medical condition files as patients depending on any machine or undergoing hemodialysis), or if MEPS had no medications listed to determine polypharmacy.

## Dependent variable

The dependent variables used in this study were HRQoL measures. We obtained HRQoL variables from the Veterans Rand 12 (VR-12) item health survey for data from 2017 through 2019 and the short form 12 Version 2 (SF-12v2) for data pulled from 2010 through 2016. The VR-12 and SF-12v2 are self-administered questionnaires that have two components each: PCS and MCS. Each component has six separate items. To make the VR-12 and SF-12v2 compatible, AHRQ applied a bridging algorithm to the MEPS data to align the summary scores (PCS and MCS) from the VR-12 and SF-12v2 questionnaires [27]. The PCS and MCS each have a mean summary score of 50 and a standard deviation of 10 in the US population. Higher scores indicate better physical and mental quality of life. The PCS provides greater weight to questions involving role limitations with physical functioning, physical health, pain, and general health.

The MCS provides greater weight to questions involving interference with emotional problems, social functioning, and mental health.

## Independent variable

The primary independent variable of interest for this study was polypharmacy. To date, there is no universal definition for polypharmacy. Some studies utilized a threshold of 5 or more concomitant medications over a certain period to define polypharmacy, while others employed a different threshold [28]. In this study, we classified polypharmacy based on the number of unique therapeutic classes of prescribed medications taken concomitantly over a specific round (e.g., 6-month data collection period with each yearly released data files designed by AHRQ such that patients in the current year are independent of patients in the previous year with each patient assigned unique weights, and variance estimation strata). We characterized polypharmacy as follows: minor polypharmacy ($\leq 4$ classes of medications), major polypharmacy (5–9 classes of medications), and hyperpolypharmacy ($\geq 10$ classes of medications) [28]. MEPS obtained information about prescription medication from participants and the pharmacy that dispenses these medications to the individuals. For each round, respondents were asked to name the prescription drug they were taking. The following information was obtained from the participants for each round: Type of medication, the specific medical condition for which it was prescribed, the commencement date (year and month) of usage, and the address of the dispensing pharmacy. Written consent was obtained from the individuals to contact their respective pharmacies. The written authorization forms were presented to the different pharmacy establishments. Pharmacies can either have a computerized printout of medications used by each participant or can report information using computer assisted telephone interviews. This process encompassed all survey participants who had obtained prescription medications from the specified pharmacies during a given year, and the following information was obtained: Type of prescription filled, the frequency of refills within a given round, the medical condition for which the prescription was provided, the national drug code (NDC), strength of the medication, quantity (package size and amount dispensed), and payment source. Furthermore, using Multum Lexicon database from Cerner Multum, Incorporated (http://www.multum.com/Lexicon.htm) [27], MEPS provided the prescription name of each medication and classified each prescribed medication into distinct therapeutic drug classes.

## Other independent variables

We included other independent variables in this study as explanatory variables. We included demographic variables such as age group (18 to 44, 45 to 64, and $\geq 65$ years), race (Hispanic, White, Black, and other race–Asian, Native Hawaiian/Pacific Islander, and Filipino), and marital status (married, and unmarried). Socioeconomic variables, namely insurance coverage (private, public, and uninsured), education attainment (college and more, < college, and missing = 4), and income level (low, middle, and high), were evaluated. Other variables included are census region (Northeast, Midwest, South, and West), physical activity (moderate to vigorous, no moderate to vigorous, and missing = 22), access to care (yes, no, and missing = 5), and comorbidities/risk factors such as diabetes, hypertension, arthritis and cardiovascular diseases (CVD). Access to care was used to determine if a patient had a designated location, such as a doctor's office or health care facility, where they can go to receive care if they are ill or require guidance regarding their health condition. Additionally, psychiatric conditions that include depression, severe mental disability (SMD), and anxiety were all taken into consideration. Kessler Index (K6) scale is a validated questionnaire used to determine mental conditions with

higher values indicating SMD. A dichotomous cut-point of 13 or more was applied to classify SMD, as described by Kessler et al. [29]. The Patient Health Questionnaire (PHQ-2) in MEPS was used to determine depression using a cut-point of 3 as deemed reasonable based on the recommendation by Kroenke et al. [30]. The population was further defined using the number of comorbidities to estimate the association and severity of comorbidities on HRQoL.

### Data analysis

Rao-Scott Chi-square test was applied to the categorical variables to determine significant differences in the population characteristics across degrees of polypharmacy. We utilized unadjusted ordinary least squares regression to evaluate the association between the explanatory variables and HRQoL (PCS and MCS). We performed further analysis using multivariable ordinary least squares regression to investigate the independent association between polypharmacy and HRQoL while controlling for other explanatory variables such as sex, age, race/ethnicity, marital status, income level, prescription drug coverage, health insurance coverage, physical activity, access to care, census region, education, number of comorbidities, psychiatric illnesses, and comorbidities/risk factors. We assessed the prevalence of the top 5 commonly used therapeutic classes of medications across age groups, sex, degree of polypharmacy, and HRQoL categories. In this case, we dichotomized HRQoL (low PCS, high PCS, low MCS, and high MCS) using the rank procedure in SAS. Furthermore, we examined the type of specific medication used for each of the top 5 therapeutic classes of medications.

The explanatory variables used in the adjusted model were assessed for multicollinearity using variance inflation factor (VIF), pairwise correlation coefficients, and tolerance indices (TIs). The presence of multicollinearity can lead to imprecise parameter estimates and standard errors, which in turn can affect the overall accuracy of the model [31, 32]. All variables had VIF of less than 5 with tolerance levels greater than 0.2, which implies the absence of multicollinearity [31]. All other inference assumptions of the models were examined to ensure that all requirements were met. Missing data was addressed using pairwise deletion. Given the complex sampling design of the MEPS data, all statistical analyses were performed utilizing strata, clustering, and SAQ weight in testing for statistical differences with a significant level (alpha) of 0.05. All analyses were conducted using SAS software, version 9.4 (SAS institute. Cary, NC). The datasets used in this study were extracted in January 2022, the analyses were started right after and completed in August of 2022. This study was considered exempt status by West Virginia University since the MEPS data is a publicly available national representative dataset with deidentified individuals.

### Results

Among 339,883 participants in the MEPS data, 932 had CKD, of which 649 met the inclusion criteria (Fig 1). The mean age and standard deviation of CKD patients was 61.55 ± 13.93. Sex was not significantly different across the levels of polypharmacy. Over two-thirds (76.89%) of participants exhibited major polypharmacy or hyperpolypharmacy. CKD patients were predominantly White (62.89%), followed by Blacks (20.66%), Hispanics (10.32%), and other races (6.14%). The proportion of patients with major polypharmacy (49.20%) and hyperpolypharmacy (84.28%) was more pronounced when the number of comorbidities were ≥ 3. All psychiatric illnesses (i.e., depression, SMD, and anxiety) and comorbidities (i.e., diabetes, hypertension, arthritis, and CVD) were significantly different (p-value < 0.001) across all degrees of polypharmacy. The most prevalent comorbidity among patients with CKD was hypertension (84.82%). Across psychiatric illnesses, hyperpolypharmacy occurred more frequently in patients with depression (45.30%), SMD (50.43%), and anxiety (54.07%) (Table 1).

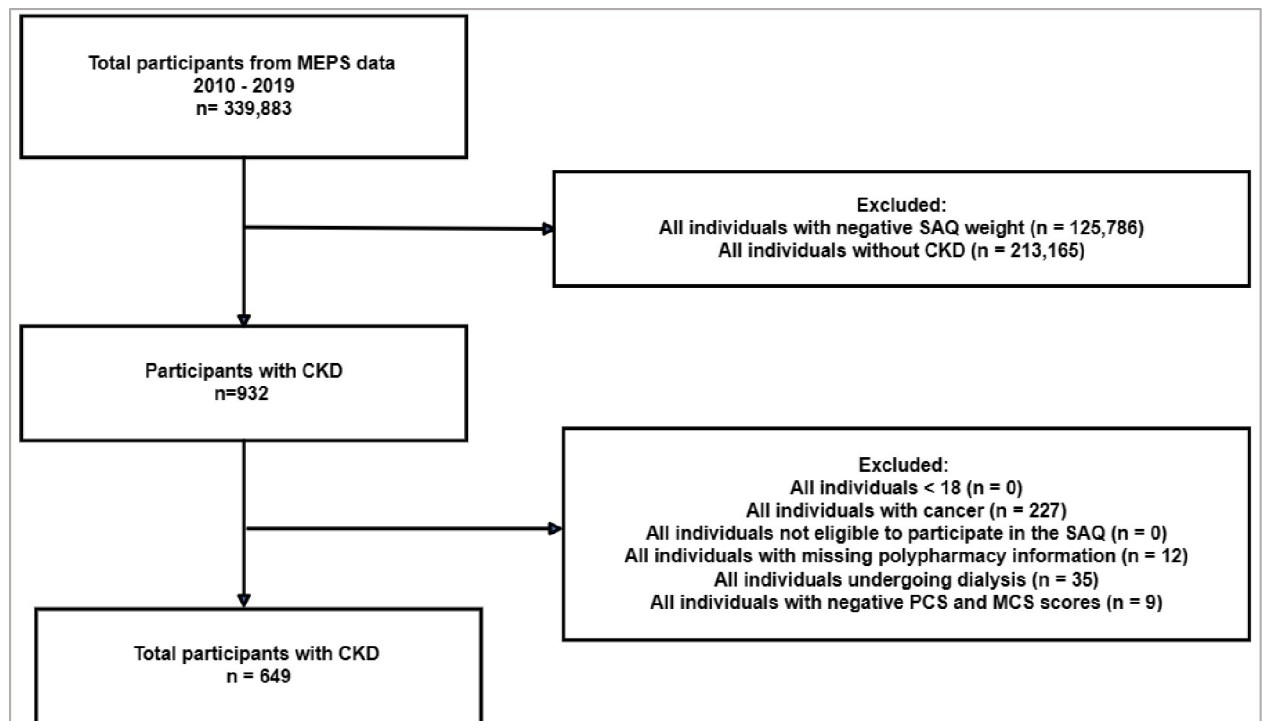

**Fig 1. Study flow diagram.** Self-administered questionnaire (SAQ), Chronic kidney disease (CKD), Medical expenditure panel survey (MEPS), Physical component summary (PCS), Mental component summary (MCS).

### The top 5 commonly used medications among non-dialysis CKD patients

The top 5 therapeutic classes of medications taken by non-dialysis CKD patients are antihyperlipidemic agents (56.31%), beta-adrenergic blocking agents (49.71%), antidiabetic agents (42.17%), analgesics (42.14%), and diuretics (39.65%). Antihyperlipidemic and beta-adrenergic blocking agents were used across all groups (sex, age group, degree of polypharmacy, and HRQoL categories), and patients with low PCS and hyperpolypharmacy used a higher proportion of these two classes of medications. All groups used analgesics except for patients 65 years and older, and all groups used antidiabetic agents except those between 18 and 44 years and patients with minor polypharmacy. Diuretics, on the other hand, were used by males, patients with hyperpolypharmacy, patients with low PCS and MCS, high MCS, and patients 65 years and older (Fig 2). Of the individuals taking analgesics drug class, acetaminophen-hydrocodone (28.01%) and tramadol (15.68%) were the number 1 and 2 medications taken by this subgroup. Additionally, among individuals taking diuretics, furosemide (60.07%) and hydrochlorothiazide (21.60%) were the number 1 and 2 medications in this class (Fig 3). Furthermore, for each therapeutic drug class, except for atorvastatin and simvastatin, within the antihyperlipidemic agent, the proportion of prescription medications not covered by insurance exceeded those covered by insurance (S1 Fig).

### Unadjusted model of the association between polypharmacy and health-related quality of life

Patients with hyperpolypharmacy had the lowest mean PCS at 29.62 (95% CI: 27.30–31.94), which was followed by major polypharmacy at 34.90 (95% CI: 32.68–37.12) and patients with

**Table 1. Characteristics of study participants by degrees of polypharmacy (Weighted N = 667,989).**

| Parameters | N = 649 | Minor Polypharmacy<br>n = 150 (22.27%) | Major Polypharmacy<br>n = 307 (48.24%) | Hyperpolypharmacy<br>n = 192 (29.48%) | p-value |
|---|---|---|---|---|---|
| | No. (W%) | No. (W%) | No. (W%) | No. (W %) | |
| **Sex** | | | | | 0.111 |
| Male | 306 (50.29) | 73 (24.49) | 152 (49.51) | 81 (25.99) | |
| Female | 343 (49.71) | 77 (20.02) | 155 (46.96) | 111 (33.02) | |
| **Age** | | | | | |
| 18–44 | 77 (12.59) | 28 (38.77) | 33 (47.97) | 16 (13.26) | <0.001 |
| 45–64 | 283 (41.16) | 68 (23.09) | 120 (41.97) | 95 (34.94) | |
| ≥ 65 | 289 (46.25) | 54 (17.05) | 154 (53.90) | 81 (29.05) | |
| **Race/Ethnicity** | | | | | 0.392 |
| White | 289 (62.89) | 66 (21.25) | 132 (46.88) | 91 (31.87) | |
| African American Black | 211 (20.66) | 44 (21.17) | 108 (54.30) | 59 (24.54) | |
| Hispanic | 102 (10.32) | 29 (28.26) | 44 (45.85) | 29 (25.89) | |
| Other | 47 (6.14) | 11 (26.39) | 23 (45.89) | 13 (27.72) | |
| **Marital Status** | | | | | 0.897 |
| Married | 287 (52.84) | 64 (21.59) | 140 (49.08) | 83 (29.33) | |
| Unmarried | 362 (47.16) | 86 (23.04) | 167 (47.30) | 109 (29.66) | |
| **Income Level** | | | | | 0.291 |
| Low Income | 343 (44.31) | 67 (20.01) | 163 (46.52) | 113 (33.47) | |
| Middle Income | 159 (25.67) | 46 (25.27) | 74 (50.38) | 39 (24.35) | |
| High Income | 147 (30.02) | 37 (23.04) | 70 (48.96) | 40 (28.00) | |
| **Prescription Drug Coverage** | | | | | 0.495 |
| Yes | 214 (42.44) | 59 (24.04) | 99 (48.74) | 56 (27.22) | |
| No | 435 (57.56) | 91 (20.96) | 208 (47.88) | 136 (31.15) | |
| **Health Insurance Coverage** | | | | | 0.338 |
| Private | 271 (51.02) | 72 (23.71) | 128 (47.95) | 71 (28.34) | |
| Public | 365 (45.53) | 72 (19.23) | 174 (50.83) | 119 (29.94) | |
| Uninsured | 13 (3.44) | 6 (41.15) | 5 (18.44) | 2 (40.41) | |
| **Physical Activity** | | | | | <0.001 |
| Moderate to Vigorous | 178 (31.38) | 57 (34.68) | 84 (46.97) | 37 (18.34) | |
| No Moderate to Vigorous | 449 (68.62) | 83 (15.68) | 217 (49.50) | 149 (34.82) | |
| **Access to Care** | | | | | <0.001 |
| Yes | 596 (93.00) | 128 (20.40) | 283 (48.45) | 185 (31.15) | |
| No | 48 (7.00) | 21 (46.45) | 20 (43.62) | 7 (9.93) | |
| **Census Region** | | | | | 0.063 |
| Northeast | 101 (17.00) | 23 (18.61) | 46 (42.92) | 32 (38.47) | |
| Midwest | 140 (21.58) | 30 (20.11) | 63 (50.31) | 47 (29.58) | |
| South | 274 (41.34) | 63 (25.51) | 138 (52.15) | 73 (26.34) | |
| West | 134 (20.08) | 34 (29.26) | 60 (42.48) | 40 (28.25) | |
| **Education** | | | | | 0.710 |
| College and more | 217 (40.51) | 55 (21.02) | 104 (48.88) | 58 (30.10) | |
| < College | 428 (59.49) | 95 (23.52) | 201 (48.26) | 132 (28.22) | |
| **Number of Comorbidities** | | | | | <0.001 |
| 0 | 28 (4.84) | 22 (71.29) | 5 (23.79) | 1 (4.93) | |
| 1 | 112 (17.43) | 50 (47.81) | 52 (46.32) | 10 (5.88) | |
| 2 | 165 (25.93) | 49 (28.04) | 90 (58.95) | 26 (13.01) | |
| ≥ 3 | 344 (51.80) | 29 (6.21) | 160 (45.82) | 155 (47.97) | |

*(Continued)*

**Table 1.** (Continued)

| Parameters | N = 649 | Minor Polypharmacy | Major Polypharmacy | Hyperpolypharmacy | p-value |
|---|---|---|---|---|---|
| | | n = 150 (22.27%) | n = 307 (48.24%) | n = 192 (29.48%) | |
| | No. (W%) | No. (W%) | No. (W%) | No. (W %) | |
| **Psychiatric Illness** | | | | | |
| Depression | 136 (19.75) | 21 (12.38) | 62 (42.33) | 53 (45.30) | <0.001 |
| Severe Mental Disability | 82 (12.21) | 15 (11.11) | 32 (38.46) | 35 (50.43) | <0.001 |
| Anxiety | 107 (18.73) | 12 (9.68) | 38 (36.25) | 57 (54.07) | <0.001 |
| **Comorbidities** | | | | | |
| Diabetes | 348 (52.42) | 46 (10.80) | 165 (47.12) | 137 (42.07) | <0.001 |
| Hypertension | 560 (84.82) | 107 (18.73) | 277 (49.37) | 176 (31.90) | <0.001 |
| Arthritis | 283 (44.46) | 43 (13.32) | 126 (44.50) | 114 (42.18) | <0.001 |
| CVD | 279 (40.37) | 26 (8.70) | 129 (45.25) | 124 (46.05) | <0.001 |

Variables were analyzed using Rao-Scott Chi-square test to observe differences in proportions across the degrees of polypharmacy. Minor polypharmacy (0–4 classes of medications), Major Polypharmacy (5–9 classes of medications) and hyperpolypharmacy ($\geq$ 10 classes of medications). W: Weighted; No.: Number; Income level was based on the classification of federal poverty level (FPL), with low income ($\geq$100 –<200% FPL), middle income ($\geq$ 200%–<400% FPL), and high income ($\geq$ 400% FPL), cardiovascular disease (CVD). Additional information about how weights are calculated can be found in the MEPS website (https://meps.ahrq.gov/mepsweb/).

minor polypharmacy had the highest PCS at 41.46 (95% CI: 38.89–44.03) with a p-value of <0.0001. Age group, income level, prescription drug coverage, health insurance, physical activity, access to care, number of comorbidities, psychiatric illnesses, diabetes, and arthritis were all significantly associated with PCS based on a significance level of 0.05. CKD patients with depression and SMD had significantly lower PCS than those without (Table 2). Subsequently, CKD patients with hyperpolypharmacy had the lowest MCS at 44.69 (95% CI: 41.30–48.09), which was followed by patients with major polypharmacy at 49.39 (95% CI: 47.91–50.8) and the highest values were observed among patients with minor polypharmacy 51.01 (95% CI: 49.22–52.76) with a p-value of 0.005. Age group, marital status, income level, health insurance coverage, prescription drug coverage, physical activity, number of comorbidities, psychiatric illnesses, diabetes, and arthritis were all significantly associated with MCS based on a significant level of 0.05. CKD patients with psychiatric illnesses had significantly lower MCS than those without.

## Adjusted model of the association between polypharmacy and health-related quality of life (Physical component summary)

In the adjusted model, the mean PCS was significantly lower (by 3 points) among CKD patients with major polypharmacy when compared with minor polypharmacy [Beta = -3.12, (95% CI: -3.62, -2.66), p-value <0.001]. Similarly, the PCS was four points lower among CKD patients with hyperpolypharmacy compared to those with minor polypharmacy [Beta = -4.56, (95% CI: -5.11, -4.01), p-value <0.001]. Based on the parameter estimates and p-values, other factors that are independently and negatively (Beta $\leq$ -1) associated with HRQoL are age group, income level, health insurance coverage, physical activity, census region, number of comorbidities, depression, diabetes, arthritis, and CVD. Among these, patients with depression [Beta = -6.11 (95% CI: -6.68, -5.54), p-value <0.001], patients with no moderate to vigorous physical activity [Beta = -4.91 (95% CI: -5.37, -4.46), p-value <0.001] and patients with public insurance [Beta = -4.70 (95% CI: -5.14, -4.26), p-value < 0.001] had the strongest association with HRQoL in comparison to patients without the listed attributes (Table 3).

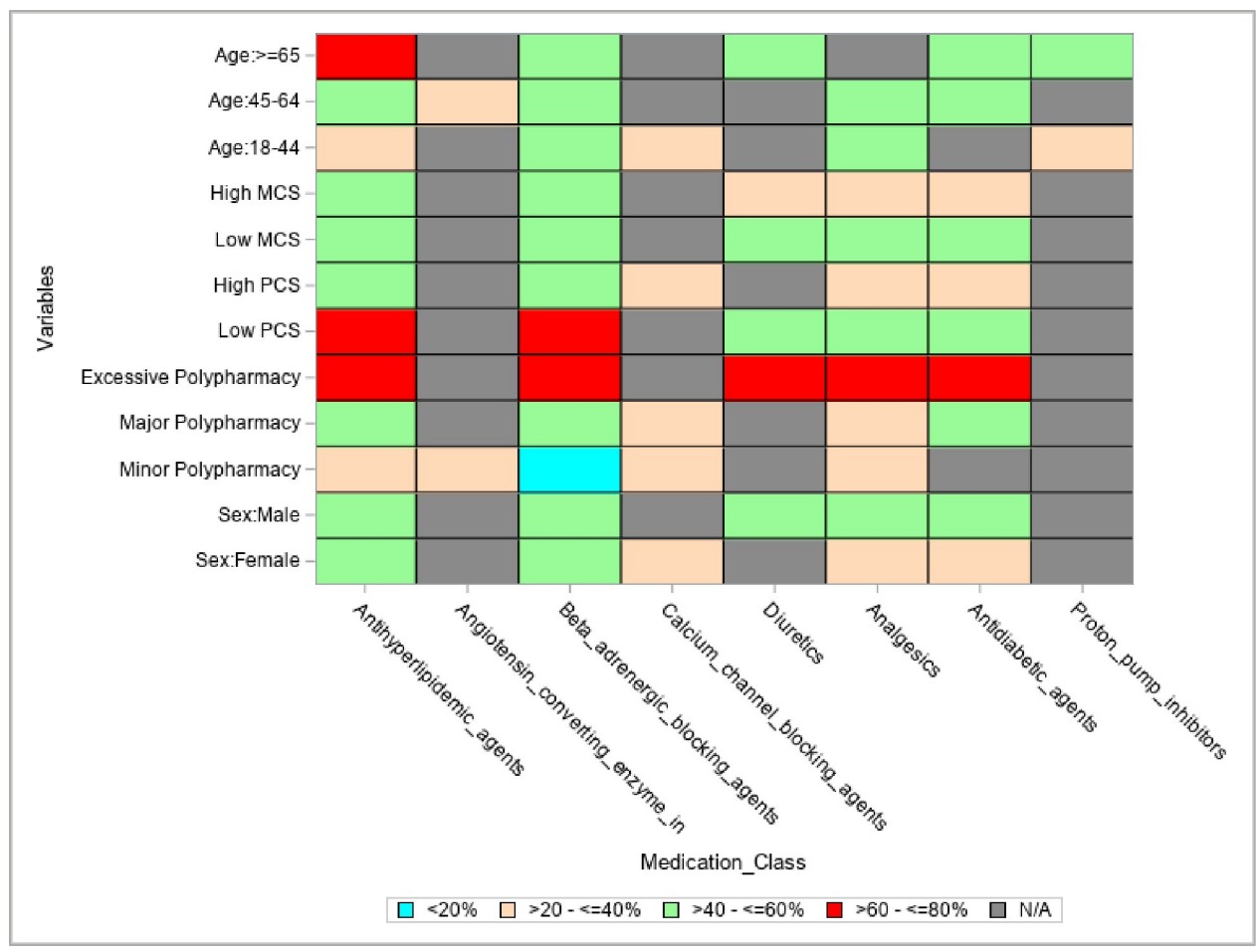

**Fig 2. Top 5 therapeutic classes of medications among non-dialysis CKD patients by age, HRQoL, polypharmacy, and sex.** The gray cells for each row or column should not be counted; percentage (%) = weighted percentage; low PCS (≤31.78), high PCS (>31.78–63.34), low MCS (≤49.51), and high MCS (> 49.51–71.66); Excessive Polypharmacy ≥ 10, Major Polypharmacy 5–9, Minor Polypharmacy < 5, HRQoL: Health-related quality of life, CKD: chronic kidney disease, Angiotensin_converting_enzyme_in: Angiotensin converting enzyme inhibitors.

### Association between polypharmacy and health-related quality of life (Mental component summary)

After controlling for all other explanatory variables in the model, major polypharmacy and hyperpolypharmacy were significantly and negatively associated with lower MCS when compared to patients with minor polypharmacy [Beta = -0.38 (95% CI: -0.55, -0.20), p-value <0.001 and Beta = -1.70 (95% CI: -2.01, -1.40), p-value <0.001]. All other explanatory variables that were independently and negatively (Beta ≤ -1) associated with lower MCS are race (African American black when compared with whites), income level, prescription drug coverage, physical activity, number of comorbidities, and psychiatric illnesses. Of these, three factors had the strongest association with MCS. They were as follows, depression [with vs. without Beta = -10.57 (95% CI: -11.17, -9.97), p-value < 0.001], SMD [with vs. without Beta = -8.93 (95% CI: -9.42, -8.43) p-value < 0.001] and patients with three or more comorbidities [with vs. without Beta = -4.63 (95% CI: -5.97, -3.30), p-value <0.001] (Table 3).

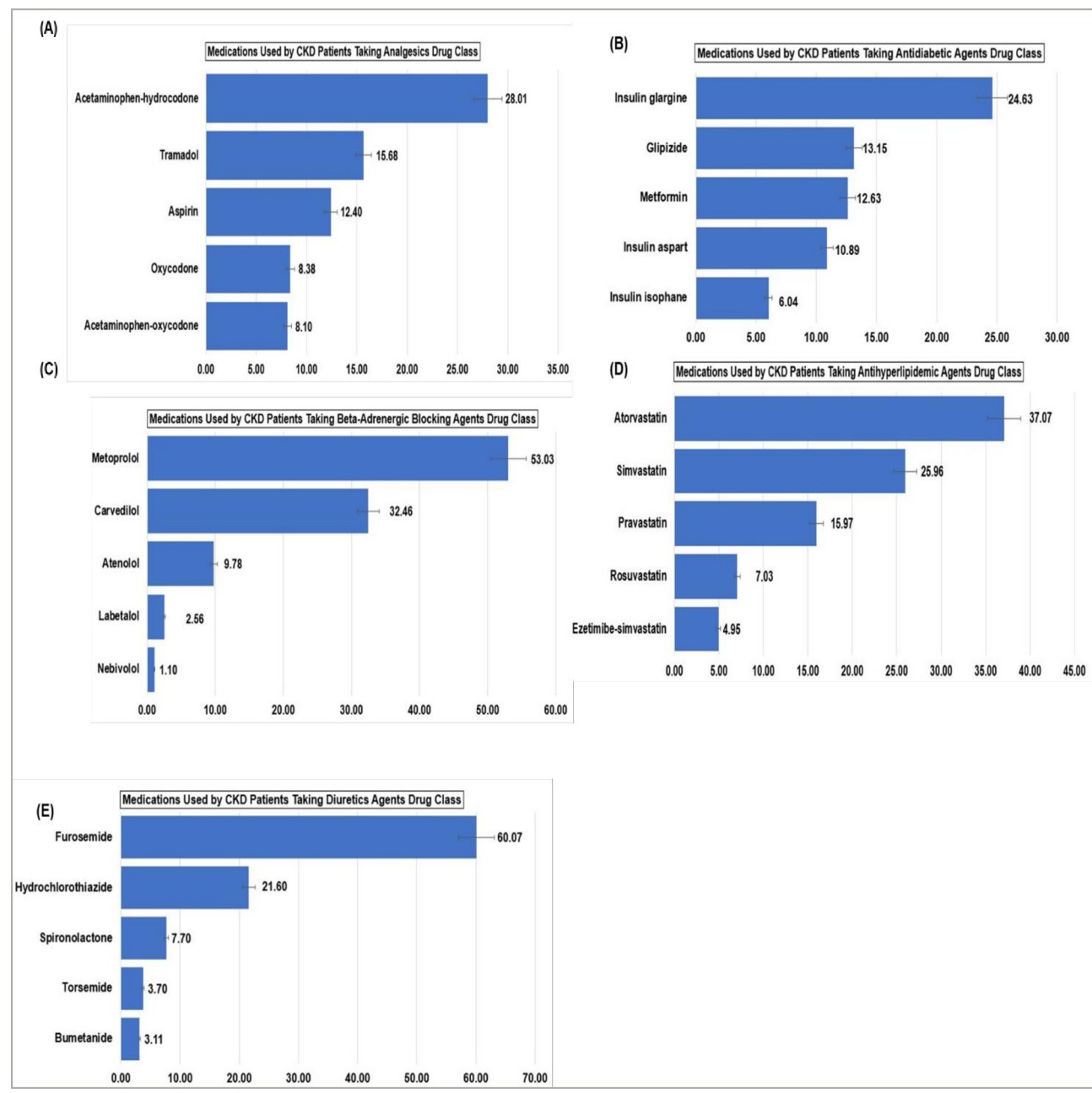

**Fig 3. Top 5 medications used by non-dialysis CKD patients by therapeutic drug class (percentage (%) = weighted percentage).**

## Discussion

The current study examined the association between polypharmacy and HRQoL among non-dialysis CKD patients. The results showed that polypharmacy is negatively associated with HRQoL. We found that major polypharmacy and hyperpolypharmacy were associated with lower PCS by 3 and 4 points compared to patients with minor polypharmacy. Our study also revealed that major and hyperpolypharmacy were associated with lower MCS by 0.37 and 1 point compared to patients with minor polypharmacy. As shown by Samsa et al., a clinically significant difference in HRQoL is between 3 and 5 [33]; as such, the result of our study implies

**Table 2.  Unadjusted ordinary least squares regression with 95% confidence interval (physical component summary and mental component summary).**

| Characteristics | PCS | | MCS | |
|---|---|---|---|---|
| | Mean (95% CI) | p-value | Mean (95% CI) | p-value |
| **Polypharmacy** | | <0.001 | | 0.005 |
| Minor polypharmacy | 41.46 (38.89–44.03) | | 51.01 (49.22–52.79) | |
| Major Polypharmacy | 34.90 (32.68–37.12) | | 49.39 (47.91–50.88) | |
| Hyperpolypharmacy | 29.62 (27.30–31.94) | | 44.69 (41.30–48.09) | |
| **Sex** | | 0.227 | | 0.130 |
| Male | 35.63 (33.59–37.67) | | 49.47 (47.85–51.10) | |
| Female | 33.97 (31.83–36.10) | | 47.24 (44.87–49.62) | |
| **Age Group** | | <0.001 | | 0.015 |
| 18–44 | 41.74 (38.01–45.48) | | 49.54 (46.55–52.52) | |
| 45–64 | 34.80 (32.36–37.23) | | 45.98 (43.48–48.48) | |
| $\geq$ 65 | 32.92 (30.80–35.04) | | 50.17 (48.57–51.78) | |
| **Race/Ethnicity** | | 0.268 | | 0.062 |
| White | 34.17 (31.83–36.50) | | 49.10 (46.87–51.32) | |
| African American Black | 37.10 (34.74–39.47) | | 48.19 (46.45–49.93) | |
| Hispanic | 34.63 (30.94–38.32) | | 46.33 (43.63–49.04) | |
| Other | 33.85 (29.30–38.41) | | 44.92 (42.27–47.57) | |
| **Marital Status** | | 0.315 | | 0.041 |
| Married | 35.48 (33.28–37.67) | | 49.75 (48.09–51.40) | |
| Unmarried | 34.05 (32.04–36.06) | | 46.82 (44.50–49.14) | |
| **Income Level** | | <0.001 | | <0.001 |
| High Income | 38.58 (35.90–41.27) | | 51.94 (50.16–53.71) | |
| Middle Income | 36.13 (33.78–38.49) | | 48.87 (46.29–51.46) | |
| Low Income | 31.47 (29.51–33.43) | | 45.66 (43.70–47.61) | |
| **Prescription Drug Coverage** | | <0.001 | | 0.007 |
| Yes | 38.15 (35.74–40.55) | | 50.50 (48.79–52.22) | |
| No | 32.34 (30.53–34.15) | | 46.79 (44.72–48.87) | |
| **Health Insurance Coverage** | | <0.001 | | 0.003 |
| Private | 37.89 (35.74–40.04) | | 50.55 (48.91–52.19) | |
| Public | 31.29 (29.51–33.07) | | 46.44 (44.91–47.98) | |
| Uninsured | 35.50 (25.82–45.18) | | 41.44 (19.37–63.50) | |
| **Physical Activity** | | <0.001 | | <0.001 |
| Moderate to Vigorous | 41.55 (41.28–41.83) | | 51.81 (51.69–51.93) | |
| No Moderate to Vigorous | 32.41 (32.22–32.60) | | 47.14 (46.85–47.44) | |
| **Access to Care** | | 0.007 | | 0.117 |
| Yes | 34.53 (33.55–35.51) | | 48.27 (47.03–49.50) | |
| No | 38.77 (35.95–41.59) | | 49.87 (48.31–51.43) | |
| **Census Region** | | 0.636 | | 0.186 |
| Northeast | 35.87 (32.26–39.47) | | 48.99 (45.73–52.26) | |
| Midwest | 34.98 (31.52–38.44) | | 49.44 (47.09–51.79) | |
| South | 33.66 (31.21–36.11) | | 48.63 (45.89–51.37) | |
| West | 36.06 (32.59–39.53) | | 46.14 (44.01–48.28) | |
| | PCS | | MCS | |
| Parameters | Mean (95% CI) | p-value | Mean (95% CI) | p-value |
| **Education** | | 0.192 | | 0.771 |
| College and more | 36.14 (33.66–38.62) | | 48.61 (45.61–51.61) | |
| < College | 34.17 (32.34–36.01) | | 48.11 (46.72–49.50) | |

*(Continued)*

   

**Table 2.** (Continued)

| Characteristics | PCS | | MCS | |
|---|---|---|---|---|
| | Mean (95% CI) | p-value | Mean (95% CI) | p-value |
| **Number of Comorbidities** | | <0.001 | | 0.012 |
| 0 | 43.92 (38.07–49.77) | | 51.62 (46.04–57.19) | |
| 1 | 41.61 (38.54–44.69) | | 50.27 (48.41–52.13) | |
| 2 | 37.86 (35.45–40.26) | | 50.76 (49.06–52.46) | |
| ≥ 3 | 30.13 (28.00–32.27) | | 46.22 (43.87–48.58) | |
| **Psychiatric Illness** | | | | |
| Depression | 27.72 (25.18–30.27) | <0.001 | 34.61 (31.75–37.47) | <0.001 |
| Severe Mental Disability | 29.72 (26.16–33.28) | 0.003 | 31.36 (27.51–35.20) | <0.001 |
| Anxiety | 31.77 (28.69–34.84) | 0.037 | 42.03 (37.30–46.76) | 0.002 |
| **Comorbidities/Risk Factors** | | | | |
| Diabetes vs. no | 31.99 (29.89–34.09) | <0.001 | 46.47 (44.17–48.77) | 0.003 |
| Hypertension vs. no | 34.47 (32.74–36.19) | 0.274 | 48.31 (46.72–49.90) | 0.846 |
| Arthritis vs. no | 30.83 (28.47–33.20) | <0.001 | 46.26 (43.69–48.83) | 0.008 |
| CVD vs. no | 30.08 (28.04–32.13) | <0.001 | 47.36 (45.58–49.14) | 0.179 |

Variables were analyzed using ordinary least squares regression to observe the association between HRQoL and each explanatory variable. Minor polypharmacy (0–4 classes of medications), Major Polypharmacy (5–9 classes of medications) and hyperpolypharmacy (≥10 classes of medications). CI: confidence Interval; PCS (physical component summary) and MCS (mental component summary); cardiovascular disease (CVD). Income level was based on the classification of federal poverty level (FPL), with low income (≥100 –<200% FPL), middle income (≥ 200%–<400% FPL), and high income (≥ 400% FPL).

that multiple medications had a clinically meaningful association with HRQoL in non-dialysis CKD patients.

To the best of our knowledge, this is the first study to investigate the association of polypharmacy and HRQoL among non-dialysis CKD patients using nationally representative survey data in the United States. The result of this study is not surprising as it aligns with previous research which has consistently shown that CKD patients are frequently prescribed multiple medications, given the degree of comorbid medical conditions. In another study, it was shown that CKD patients are often on cardiovascular medications such as angiotensin-converting enzyme inhibitors or angiotensin receptor blockers, beta-blockers, and diuretics, in addition to other blood pressure drugs like calcium channel blockers and dyslipidemia drugs, for example statins [14]. This may create an environment where patients with CKD can have potentially inappropriate prescribing (PIP) of medication. In a national study in Australia, approximately 35% ($n$ = 9926) of CKD patients had at least one PIP based on either the Cockcroft-Gault (CG) equation or CKD epidemiology collaboration equation (CKD-EPI) [34]. PIP can occur in various ways, such as inappropriately adjusting the medication dosage for patients with CKD [34]. It is also worth noting that both under and over prescribing can occur as part of PIP. Underprescribing of renin-angiotensin-aldosterone system (RAAS) inhibitors, statins, and active vitamin D was observed among patients with higher stages of CKD. While overprescribing of non-steroidal anti-inflammatory drugs (NSAIDS) was observed among patients with earlier stages of CKD [35]. In a Dutch study, among patients with dialysis-dependent CKD, a clinically significant difference was observed in PCS scores when patients took more than five medications and in MCS scores when patients took between 14–27 medications [26]. These findings are similar to the observations found in our study. Our study, however, found a weaker association between MCS and the number of medications prescribed. This could suggest that dialysis patients may experience a more pronounced decline in mental HRQoL than non-dialysis CKD patients.

**Table 3. Multivariable ordinary least squares regression in the association between polypharmacy and health-related quality of life.**

| | PCS | | MCS | |
|---|---|---|---|---|
| Parameter | Beta (CI) | p-value | Beta (CI) | p-value |
| **Polypharmacy** | | | | |
| Minor polypharmacy | Reference | | Reference | |
| Major polypharmacy | -3.12 (-3.62, -2.62) | <0.001 | -0.38 (-0.55, -0.20) | <0.001 |
| Hyperpolypharmacy | -4.13 (-4.74, -3.52) | <0.001 | -1.70 (-2.01, -1.40) | <0.001 |
| **Sex** | | | | |
| Male | Reference | | Reference | |
| Female | 0.79 (0.57, 1.01) | <0.001 | 0.25 (-0.24, 0.75) | 0.313 |
| **Age Group** | | | | |
| 18–44 | Reference | | Reference | |
| 45–64 | -3.35 (-3.73, -2.97) | <0.001 | -0.18 (-0.66, 0.29) | 0.443 |
| $\geq 65$ | -3.49 (-3.94, -3.04) | <0.001 | 2.26 (1.62, 2.89) | <0.001 |
| **Race/Ethnicity** | | | | |
| White | Reference | | Reference | |
| African American Black | 3.07 (2.44, 3.70) | <0.001 | -1.08 (-2.06, -0.10) | 0.030 |
| Hispanic | 0.19 (-0.78, 1.17) | 0.697 | -0.24 (-0.64, 0.17) | 0.251 |
| Other | 2.05 (1.24, 2.87) | <0.001 | -0.28 (-0.88, 0.32) | 0.355 |
| **Marital Status** | | | | |
| Married | Reference | | Reference | |
| Unmarried | -0.66 (-1.20, -0.13) | 0.016 | -0.66 (-1.17, -0.15) | 0.011 |
| **Income Level** | | | | |
| High Income | Reference | | Reference | |
| Middle Income | -2.33 (-3.24, -1.42) | <0.001 | -1.26 (-2.19, -0.32) | 0.009 |
| Low Income | -3.09 (-3.69, -2.48) | <0.001 | -2.07 (-2.61, -1.52) | <0.001 |
| **Prescription Drug Coverage** | | | | |
| Yes | Reference | | Reference | |
| No | 0.08 (-0.40, 0.55) | 0.754 | -1.67 (-2.57, -0.77) | <0.001 |
| **Health Insurance Coverage** | | | | |
| Private | Reference | | Reference | |
| Public | -4.70 (-5.14, -4.26) | <0.001 | 0.21 (-0.05, 0.48) | 0.111 |
| Uninsured | -2.42 (-2.87, -1.98) | <0.001 | 1.23 (0.43, 2.04) | 0.003 |
| **Physical Activity** | | | | |
| Moderate to Vigorous | Reference | | Reference | |
| No Moderate to Vigorous | -4.91 (-5.37, -4.46) | <0.001 | -1.73 (-2.27, -1.19) | <0.001 |
| **Access to Care** | | | | |
| Yes | Reference | | Reference | |
| No | 0.60 (0.30, 0.90) | <0.001 | -0.48 (-0.85, -0.11) | 0.011 |
| **Census Region** | | | | |
| Northeast | Reference | | Reference | |
| Midwest | -2.00 (-2.70, -1.30) | <0.001 | 1.24 (-0.16, 2.64) | 0.084 |
| South | -2.98 (-3.74, -2.22) | <0.001 | -0.22 (-1.58, 1.14) | 0.754 |
| West | -1.14 (-1.90, -0.38) | 0.004 | -2.32 (-3.65, -0.99) | <0.001 |
| | PCS | | MCS | |
| Education | Beta (CI) | p-value | Beta (CI) | p-value |
| College and more | Reference | | Reference | |
| < College | 0.05 (-0.50, 0.59) | 0.860 | -0.35 (-0.98, 0.29) | 0.286 |
| **Number of Comorbidities** | | | | |

*(Continued)*

**Table 3.** (Continued)

| Parameter | PCS | | MCS | |
|---|---|---|---|---|
| | Beta (CI) | p-value | Beta (CI) | p-value |
| 0 | Reference | | Reference | |
| 1 | -1.06 (-1.70, -0.43) | 0.001 | -3.38 (-4.11, -2.65) | <0.001 |
| 2 | -2.46 (-3.10, -1.82) | <0.001 | -3.20 (-4.30, -2.10) | <0.001 |
| $\geq$ 3 | -4.57 (-5.45, -3.68) | <0.001 | -4.63 (-5.97, -3.30) | <0.001 |
| **Psychiatric Illness** | | | | |
| Depression vs. no | -6.11 (-6.68, -5.54) | <0.001 | -10.57 (-11.17, -9.97) | <0.001 |
| Severe Mental Disability vs no | 3.67 (3.00, 4.35) | <0.001 | -8.93 (-9.42, -8.43) | <0.001 |
| Anxiety vs no | -0.51 (-0.94, -0.09) | 0.018 | -2.28 (-2.63, -1.93) | <0.001 |
| **Comorbidities/Risk Factors** | | | | |
| Diabetes vs. no | -1.35 (-1.94, -0.75) | <0.001 | 0.04 (-0.47, 0.56) | 0.866 |
| Hypertension vs. no | 2.84 (2.39, 3.28) | <0.001 | 1.52 (1.05, 1.99) | <0.001 |
| Arthritis vs. no | -2.13 (-2.84, -1.43) | <0.001 | -0.32 (-0.76, -0.13) | 0.160 |
| CVD vs. no | -2.65 (-3.10, -2.20) | <0.001 | 0.72 (-0.03, 1.48) | 0.061 |

Beta (Parameter estimate); CI (confidence interval); PCS (physical component summary); MCS (mental component summary); CVD (cardiovascular disease); Income level was based on the classification of federal poverty level (FPL), with low income ($\geq$100 Reference <200% FPL), middle income ($\geq$ 200% Reference <400% FPL), and high income ($\geq$ 400% FPL), R-Squared for the PCS model = 35.64%, R-Squared for the MCS model = 49.62%.

Additionally, we observed that 76.89% of CKD patients reported having either major polypharmacy or hyperpolypharmacy. We found comparable results in a Japanese study that found 75% of CKD patients who were not on dialysis used five or more prescription medications excluding over-the-counter medications [12]. Our study observed a significantly lower PCS associated with depression by 6 and 4 points in patients with three or more comorbidities. Likewise, a significantly lower MCS was associated with depression by 10 points and 8 points in patients with severe mental disability. Our study found that individuals with hyperpolypharmacy had the highest prevalence of depression, SMD, and anxiety, with 84.28% of them having three or more comorbidities. It has been shown that CKD is associated with a wide range of mental (schizophrenia or bipolar affective disorder, depression, and learning disabilities) and physical (hypertension, heart failure, diabetes, coronary heart disease, and peripheral vascular disease) conditions [24]. These can subsequently affect mental and physical HRQoL among CKD patients [24]. It is noteworthy to mention that depression had the strongest association with HRQoL for both PCS and MCS components. This is a significant and clinically relevant finding, as most research in this field has primarily centered on patients receiving dialysis treatments or ESRD patients. MEPS data for this study indicated that less than 1% of the US population had CKD when the prevalence of CKD is 14% in the US. According to the CDC, nine out of 10 adults do not know they have CKD. Since MEPS data is self-reported without the use of laboratory testing, patients who reported CKD are those diagnosed or told by their healthcare professional that they have CKD. Subsequently 10% of this 14% would be much closer to the population of CKD patients observed in the MEPS data. However, since MEPS utilized a complex sampling design made to be nationally representative, our analysis was adjusted based on the weights and variance estimation provided by MEPS.

In our study, we further examined the type of therapeutic classes of medications commonly prescribed for non-dialysis CKD patients. We observed that the top 5 classes of medication commonly prescribed for non-dialysis CKD patients are antihyperlipidemic agents, beta-adrenergic blocking agents, antidiabetic agents, analgesics, and diuretics. Antihyperlipidemic

and beta-adrenergic blocking agent were among the top 5 therapeutic classes of medications used across all groups (sex, degree of polypharmacy, age group, MCS, and PCS categories). Among patients taking antihyperlipidemic agents, atorvastatin and simvastatin were the top two prescribed medications in this subgroup. Analgesics was among the top 5 classes of medications used by all groups as listed above, except for patients 65 years and above. Among patients taking analgesics, acetaminophen-hydrocodone and tramadol, opioid medications, were the two most used medications in this subgroup. Analgesic is a class of medication commonly prescribed to alleviate pain. However, it has been demonstrated that analgesics, such as opioids, were utilized in advanced stages of CKD for patients 65 years and older [36]. This finding contrasts with our study's result. Our investigation focused on the top 5 medications, excluding dialysis CKD patients. This exclusion may explain why analgesics are not among the top 5 medications for patients 65 years and older. Additionally, the use of opioids was found to be associated with increased risk of end stage kidney disease (ESKD) and death [36]. In our study, diuretics were used among males, patients with hyperpolypharmacy, patients with low PCS, high MCS, and patients 65 years and older. It has been shown that the combination of diuretics and renin-angiotensin-aldosterone system inhibitors was associated with an increased risk of kidney failure in non-dialysis CKD patients [12]. It was shown that poorer HRQoL is associated with higher regimen complexity in pre-dialysis patients, and nonadherence was associated with a decline in physical HRQoL over time [37]. Pill burden can impact the HRQoL of CKD patients due to its associated with inappropriate use of medication and nonadherence, which may contribute to elevated adverse health outcomes [38, 39].

This study utilized national representative survey data and controlled for a wide variety of variables that could influence HRQoL. The HRQoL instruments used are validated and well-used. However, there are limitations in our study that are worth noting. First, the design of this study is cross-sectional, thereby precluding the ability to infer causality. It remains indeterminate whether the association between low HRQoL and polypharmacy stems from the introduction of more medications, or an increase in comorbidities over time and vice versa. Second, it is possible that the HRQoL of CKD patients could be influenced by factors such as the severity of CKD or the addition of more comorbidities. However, the assessment of CKD severity was hindered since the MEPS data does not provide information on the different stages of CKD. Third, recall bias could result since MEPS is based on a self-report data collection process. The number of medications prescribed may not be accurately represented since it is likely that individuals may recall only the most recent medications prescribed. Fourth, since MEPS data only collects information on prescribed medications, our study did not include the number of medications resulting from over-the-counter use. Fifth, the glomerular filtration rate (GFR) is a standard indicator used in assessing the presence of CKD. However, given the self-reported nature of the MEPS data, laboratory results such as GFR were not evaluated.

## Conclusions

In conclusion, our study showed that major polypharamcy and hyperpolypharmacy among non-dialysis CKD patients was significantly associated with lower physical and mental HRQoL compared to patients with minor polypharmacy. The negative association between polypharmacy and physical HRQoL was much stronger than the negative association between polypharmacy and mental HRQoL. However, we observed that depression, severe mental disability, and anxiety were highest among individuals with hyperpolypharmacy compared to patients with major and minor polypharmacy. Depression had the strongest association with HRQoL for both PCS and MCS. Our study emphasizes the need for further assessment of the

combination of medications taken by non-dialysis CKD patients to reduce unnecessary and inappropriate drug combinations. Considering the impact of pill burden on the HRQoL among CKD patients and the possibility of some of these patients having hyperpolypharmacy, we recommend that healthcare providers be aware of this and review the medication list regularly for these patients. Additionally, a future prospective study is needed to further assess how pill burden affects the quality of life of non-dialysis CKD patients by evaluating both the positive and negative consequences of the prescribed medications. Future studies should also investigate the clinical implications of utilizing multiple medications beyond a certain threshold and their impact on the mental and physical health of non-dialysis CKD patients.

## Supporting information

**S1 Checklist. STROBE statement—checklist of items that should be included in reports of observational studies.**
(DOCX)

**S1 Fig. Top 5 medications used by non-dialysis CKD patients by therapeutic drug class with and without insurance coverage (percentage (%) = weighted percentage).**
(TIF)

**S1 File. Analytical data.**
(XLSX)

**S2 File. Medication data.**
(XLSX)

## Author Contributions

**Conceptualization:** Leonie Adjeroh.

**Data curation:** Leonie Adjeroh, Mohammad Ikram.

**Formal analysis:** Leonie Adjeroh.

**Investigation:** Leonie Adjeroh.

**Methodology:** Leonie Adjeroh, Mohammad A. Al-Mamun.

**Project administration:** Leonie Adjeroh, Mohammad A. Al-Mamun.

**Supervision:** Leonie Adjeroh, Mohammad A. Al-Mamun.

**Writing – original draft:** Leonie Adjeroh.

**Writing – review & editing:** Leonie Adjeroh, Todd Brothers, Khaled Shawwa, Mohammad Ikram, Mohammad A. Al-Mamun.

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
