## [Decision Letter · Decision Letter 0]

12 Jul 2023

PONE-D-23-16559The association between polypharmacy and health-related quality of life among non-dialysis chronic kidney disease patientsPLOS ONE

Dear Dr. Adjeroh,

Thank you for submitting your manuscript to PLOS ONE. After careful consideration, we feel that it has merit but does not fully meet PLOS ONE’s publication criteria as it currently stands. Therefore, we invite you to submit a revised version of the manuscript that addresses the points raised during the review process.

We look forward to receiving your revised manuscript.

Kind regards,

Ari Samaranayaka, PhD

Academic Editor

PLOS ONE

Journal Requirements:

2. Please include a separate caption for each figure in your manuscript.

**Additional Editor Comments:**

Study has important findings. But to make them stronger, the research process that leads to those findings should be clear and justified, and some of the places in the presentation needs attention to improve clarity. In those respects, I like to draw authors attention below points.

• Study haven’t assessed the “effect of” or “impact of” polypharmacy. Therefore all the instances like “effect of polypharmacy on HRQoL” or “polypharmacy lowered the HRQoL” should be revised to eliminate implied causation.

• The association between polypharmacy and HRQoL may be driven by CKD-related comorbidities and CKD severity. That was not eliminated in this study even though comorbidity count and some individual comorbidities were used as covariates in multivariable models. Therefore this need to be accepted as a limitation.

• The way data collected will influence the data structure, data quality, bias, confounding, etc. Readers will be curious about it since from the beginning. This is specifically relevant having datasets in panels, each with 5 rounds spanning over 2 years, data period is being 10 years (longitudinal?), and sampling design leading to sampling weights (abstract). No mention about the survey design except a brief mention about a complex sampling design in the data analysis section. Most questions come to readers minds like: can same people captured in data in multiple years, observations from individuals are independent of each other or not, … etc will not arise, or can easily be clarified, if the data collection is briefly explained upfront.

• Polypharmacy grouping was based on number of medications (I assume ignoring dose and frequency). If a person can be captured in data multiple times over the 10 year data period, categorizing people into those groups was based on which time point? Authors say, “In this study, we classified polypharmacy based on the number of unique therapeutic classes of prescribed medications taken concomitantly over a specific round for each year”. This says, person have to take certain number of medications in EACH YEAR to be classified into a specific polypharmacy category. Did the survey ask medication for each year in last X years?

• Abstract and table3 reported Beta, SE, P for regression coefficients. Confidence intervals for beta are more informative than SE.

• Main results, abstract, and conclusion based on statistical significance only, practical importance ignored. For example, although Pvalues are very small, Beta values (ie, difference in outcome between comparison groups) are also small , they range from -4.56 to -0.37 in adjusted model. How important these observed differences when the plausible ranges in PCS and MCS are large (0 to 100)?

• I am concerned about the conclusion “This study highlights the need for interventions or alternative treatment plans to minimize the use of multiple medications while improving the HRQoL of CKD patients”. Even though HRQoL is lower in heavier polypharmacy groups, how can we eliminate the possible claim that polypharmacy must have helped to maintain HRQoL at this level, and it is likely to become even lower if medication use is minimized in this multi-morbid population?

• Participants were selected from a large dataset that was generated using a self-administered questionnaire; ICD codes were used to identify those with CKD from that dataset. Have people self-reported their ICD codes in the questionnaire? If not from where those codes came?

• Excluded people with negative weight, negative PCS, negative MPS. How can they be negative (assume PCS and MPS were not re-scalled to Z scores)? Are you simply referring to the data cleaning process used to exclude invalid data entry in self-report? Also excluded those deceased. If the study is cross sectional (ie, data collected only once) you cannot have deceased people in self-administrated data to exclude.

• Authors have checked for the multicollinearity. Surprising that they didn’t find a presence of multicollinearity between polypharmacy and comorbidity count in that process, and also between comorbidity count and individual comorbidities.

• Should not describe non finding a statistically significant difference between groups as showing an equality (eg, “sex was equivalent to 50% of men and women”).

• Figure2 . Arrange the legend in increasing order. Title refers to top 5 med classes but 8 in the figure.

• Authors have listed most commonly used drugs in some drug classes. Can this be explained by access issue (eg; cost, covered by insurance or public funds, etc) rather than due to medical issues.

• Table3. Please indicate reference category as such, instead of using “-“.

• Authors say “…, the PCS was four times lower among CKD patients with excessive polypharmacy than those with minor polypharmacy (ꞵ = -4.56, SE =0.28, p-value <0.0001)”. This is an incorrect interpretation of OLS regression coefficient.

Also say “The results showed that polypharmacy may significantly lower the quality of life of non-dialysis CKD patients”. Please revise this to eliminate implied causation, study haven’t assessed the causation.

• Table2 footnote: what is health polypharmacy (is the sentence incomplete?).

• When presenting something in X ±Y format, please be specific about what is X and what is Y (eg; “mean age was 61.55±13.93”; here 13.93 is SD or SE or IQR, or something else?).

• Table1 & 2 footnotes. Please remove notes that are not applicable to that table (eg, CI in table1, W in table2).

• Unrealistic level of precision (ie, too many DPs) in Pvalues in tables and text.

• Replace multivariave with multivariable, this study haven’t used multivariate models.

Reviewers' comments:

Reviewer's Responses to Questions

**Comments to the Author**

1. Is the manuscript technically sound, and do the data support the conclusions?

Reviewer #1: Partly

Reviewer #2: Partly

2. Has the statistical analysis been performed appropriately and rigorously? 

Reviewer #1: I Don't Know

Reviewer #2: Yes

3. Have the authors made all data underlying the findings in their manuscript fully available?

Reviewer #1: Yes

Reviewer #2: Yes

4. Is the manuscript presented in an intelligible fashion and written in standard English?

Reviewer #1: Yes

Reviewer #2: Yes

5. Review Comments to the Author

Reviewer #1: Adjeroh et al., report an association between reduced health related quality of life (HRQoL) and polypharmacy in patients with chronic kidney disease (CKD). The paper is clearly written and addresses a relevant clinical issue. The authors state most of this work has been done in dialysis, little in CKD.

There are however several issues that should be addressed prior to consideration of publication:

1. The fundamental issues with this paper is whether the polypharmacy is the cause or a consequence of the reduced HRQoL. The authors seem to claim it is the polypharmacy that reduces the HRQoL, but this is impossible to know from this study. There is purely an association which is not surprising.

2. The patient selection and timing of the surveys and data collection must be more clearly stated. Were the medication data and HRQoL surveys done simultaneously?

3. Why were there so few patients with CKD enrolled (0.2% - when the prevalence is 15% in the population?) Were these all stage IV CKD?

4. Why were only DM, HT and arthritis considered as comorbidities? Most CKD patients with die of cardiovascular disease before needing dialysis, this would have been an important group to report on.

5. What was the definition of “access to care” used?

6. Whether the indeed number of medications “excessive” or not is also a question. Patients with CKD do have many medications, in many cases because of the associated comorbidities, but also with increasing severity of kidney disease, as one tries to stave off the need for dialysis. I am in full agreement that polypharmacy should be examined in each patient and the medications rationalized as much as possible, sometimes it is not possible. Another word should be chosen rather than “excessive”. Avoid loaded phrases like “patients being constantly prescribed…” etc…

7. Was there any way to assess stage of CKD and or number of comorbidities here, as these would be major confounders of assessment of HRQoL? The dialysis population may be more uniform in this sense.

8. Was it possible to derive the pill burden in this study? Some medications e.g. sodium bicarbonate or phosphate binders may be two class of medication but may bring 9 tablets a day each. The numbers of pills may well reduce HRQoL and some classes more than others (e.g. phosphate binders). It would have been of interest to assess I there is a relationship between drug the classes and HRQoL. Pill burden and number of medication classes cannot be used interchangeably.

9. In the tables it would make more sense to have the percentages across the horizontal than the vertical, to understand what proportion of e.g. male patients had which degree of polypharmacy (I.e. add up to 100 horizontally and not vertically)

10. The discussion should include all the components that were significant in the analyses and consider how these may have impacted/confounded the findings e.g. physical activity, health insurance etc.

Reviewer #2: This about the association between polypharmacy and quality of life in CKD makes some useful and interesting observations. Some comments are provided below for the authors to consider.

MAJOR

1. The biggest limitation of this study is its observational nature that make drawing conclusions about causation difficult, even allowing for the use of an adjusted multifactorial model. It remains likely that the higher use of the medications in patients with lower QOL may be largely explained by the fact that patients with a lower QOL have more medical problems and therefore need more medication. I don’t believe that we can easily conclude from these data that if patients with multiple health problems were prescribed fewer medications that their QOL would necessarily improve. To determine that would require a prospective and interventional study. I believe this limitation should be acknowledged in the discussion.

2. There should be some acknowledgement in the discussion that polypharmacy is not the only potentially inappropriate prescribing practice that CKD patients are at risk of. For example, Smits et al (BMJ Open, July 2019) described that both overprescribing and underprescribing were problems in CKD patients. For example, they identified underuse of RAAS inhibitors and statins in some patients who would be predicted to benefit from these medications. Another form of inappropriate prescribing in inappropriate adjustment of medication dosage in patients with CKD, as identified by Castelino et al. (BMC Nephrology, June 2020), however, the methodology in the present study by Adjeroh was unable to address this aspect, as creatinine clearance and eGFR were not assessed.

3. As per Fig 1, the rate of self-reported CKD in the patient population is much lower than would be predicted (i.e. 932/339,883 = 0.27%), considering that approximately 10% of the adult population have CKD. This observation raises questions of selection and recall bias and may influence the generalisability of the conclusions. This need to be acknowledged and discussed in the manuscript.

4. In figure 3 (top 10 medications) a mixture of generic and brand names for the medications is used. This should be corrected to only use the generic drug names. Moreover, in several cases the same drug is listed twice in a graph with the generic and brand names being listed as different drugs (e.g. 3D atorvastatin and Lipitor, rosuvastatin and Crestor, 3E furosemide and Lasix). Please address this.

MINOR

Data and Methods

1. Line 105 (Research Design) suggest “associations” rather than “effects”.

2. I have difficulty understanding this sentence (Data Analysis, line 184-186) “The missing data from the three explanatory variables (education, access to care, and physical activity) were disregarded in the model since no discernible missing pattern was observed (this was assessed using proc mi in SAS), with the overall missingness of these variables accounting for less than 5% of the eligible participants”. Please review and consider revising this sentence.

Results

1. Line 244, pg 16. I think this should be “four points” instead of “four times”. Please review.

Discussion

1. Line 277, pg 19. Suggest change “the effects of polypharmacy on the HRQoL”, to “the associations between polypharmacy and HRQoL”.

2. Line 303, pg 20. Suggest change “plausible” to relevant”.

3. Line 314, pg 21. Hydrocodone is also an opioid.

4. Line 315-317. Please review this sentence “Although, it was shown that analgesics such as opioids were used in advanced stages of CKD for patients 65 years and older [36], which contrasts with our study findings”. I am not clear what you are trying to say here.

5. Line 322. Recommend change “increased the risk” to “was associated with an increased risk”.

6. PLOS authors have the option to publish the peer review history of their article (what does this mean?). If published, this will include your full peer review and any attached files.

Reviewer #1: No

Reviewer #2: **Yes: **Peter F Mount

---

## [Author Response · Author response to Decision Letter 0]

18 Sep 2023

We extend our gratitude to the reviewers for generously dedicating their valuable time and expertise towards reviewing our manuscript. We have combined all comments and meticulously addressed each comment in accordance with the reviewers’ recommendations. The remarks from the reviewers have been formatted in italics for clarity. The changes can be seen in yellow highlights and track changes in the file “Revised Manuscript with Track Changes.”

Reviewer #1

1. Study haven’t assessed the “effect of” or “impact of” polypharmacy. Therefore, all the instances like “effect of polypharmacy on HRQoL” or “polypharmacy lowered the HRQoL” should be revised to eliminate implied causation.

o Authors response: Thank you for pointing this out. We have modified all instances of "effect of" or "impact of" with the term "association" to clarify that this study does not establish a causal relationship. Additionally, we have altered the phrase "polypharmacy lowered the HRQoL" and carefully reviewed the entire article to replace statements that implied causation with statements that indicate an association. 

2. The association between polypharmacy and HRQoL may be driven by CKD-related comorbidities and CKD severity. That was not eliminated in this study even though comorbidity count and some individual comorbidities were used as covariates in multivariable models. Therefore, this needs to be accepted as a limitation.

o Authors response: Thank you for calling our attention to this important additional information. We agree that the severity as well as comorbidities of CKD can have an association with HRQoL among CKD patients and for this reason we have included this statement in the limitation section of the manuscript since the MEPS data does not provide information on data assessing the severity of CKD (Line number 385 to 388). 

o Also note that in the manuscript, we have adjusted for multiple comorbidities such as hypertension, arthritis, anxiety, depression, CVD, and the number of comorbidities (n = 0, 1, 2, or ≥ 3) to minimize any confounding that may arise from these.

3. The way data collected will influence the data structure, data quality, bias, confounding, etc. Readers will be curious about it since from the beginning. This is specifically relevant having datasets in panels, each with 5 rounds spanning over 2 years, data period is being 10 years (longitudinal?), and sampling design leading to sampling weights (abstract). No mention about the survey design except a brief mention about a complex sampling design in the data analysis section. Most questions come to readers minds like: can same people captured in data in multiple years, observations from individuals are independent of each other or not, … etc will not arise, or can easily be clarified, if the data collection is briefly explained upfront.

o Authors response: We do agree that more information is needed to improve the understanding of data collection process and as a result a detailed description of the data has been included in the data source section of the manuscript (Line number 108 to 135). MEPS consolidates its datasets in panels, each consisting of 5 rounds spanning over two years. Interviews are conducted every six months through 30 months, and data is released each year. MEPS HC data file is released as a full-year consolidated data file with associated weights and estimates or as a longitudinal file with weights and estimates for the combined two years data collect process. This study utilized the yearly consolidated data files and the accompanied SAQ weights and variance estimation strata. The 10 years data was included to increase sample size for the CKD population. More information about the data can be obtained through the link (https://www.ahrq.gov/data/meps.html) which is also referenced in the manuscript. 

o We chose not to employ the longitudinal dataset that covered the two-year data collection period due to the significantly low sample size observed during the follow-up for participants with CKD. In other words, the data utilized in this study are from independent yearly released datasets, and individuals were not tracked from 2010 through 2019. 

4. Polypharmacy grouping was based on number of medications (I assume ignoring dose and frequency). If a person can be captured in data multiple times over the 10 year data period, categorizing people into those groups was based on which time point? Authors say, “In this study, we classified polypharmacy based on the number of unique therapeutic classes of prescribed medications taken concomitantly over a specific round for each year”. This says, person have to take certain number of medications in EACH YEAR to be classified into a specific polypharmacy category. Did the survey ask medication for each year in last X years?

o Authors response: Polypharmacy information was based on the number of unique therapeutic classes of prescribed medications utilized by participants in each round of data collection that is conducted every 6 months. MEPS obtained written consent from individuals to contact their respective pharmacies. The written authorization forms were presented to the different pharmacy establishments. This process encompassed all survey participants who had obtained prescription medications from the specified pharmacies during a given year. The following information was obtained: Name, dosage, and quantity of each prescription filled and the frequency of refills within a given round, the medical condition for which the prescription was provided, the national drug code (NDC), strength of the medication, quantity (package size and amount dispensed), and payment source. The type of medication utilized was further reclassified by MEPS into distinct therapeutic subclasses of medications using the Multum Lexicon database from Cerner Multum (http://www.multum.com/Lexicon.htm) [1]. 

o Detailed information on how polypharmacy was obtained is included in the subsection of the manuscript titled “Independent Variable” (Line number 168 to 187).

 Reference

1. Medical Expenditure Panel Survey (MEPS) | Agency for Healthcare Research and Quality. Available from: https://www.ahrq.gov/data/meps.html

5. Abstract and table 3 reported Beta, SE, P for regression coefficients. Confidence intervals for beta are more informative than SE.

o Authors Response: We agree with the reviewer that the use of confidence intervals (CI) would be appropriate in this scenario. As such, all SEs in table 3, the abstract and the result section have been replaced with Confidence Intervals (CIs).

6. Main results, abstract, and conclusion based on statistical significance only, practical importance ignored. For example, although P-values are very small, Beta values (ie, difference in outcome between comparison groups are also small, they range from -4.56 to -0.37 in adjusted model. How important these observed differences when the plausible ranges in PCS and MCS are large (0 to 100)?

o Authors response: We thank the reviewer for this insightful comment. In a previously published article, Samsa et al. [1] provided that a clinically significant difference in HRQoL exists within the range of 3 to 5 (provided in the first paragraph of the discussion section). In our study, deviations spanning from 3 to 5 and beyond would be considered both statistically and clinically significant. Small differences in HRQoL can be important for patients dealing with chronic conditions, especially when compared with individuals without such conditions, and this effect is particularly pronounced for patients with comorbid conditions [2]. 

o Existing research has demonstrated a consistent decline in HRQoL correlating with an increasing count of chronic conditions [2]. Our current study revealed that a notable 78% of CKD patients exhibited the presence of 2 or more comorbidities. 

References

1. Samsa G, Edelman D, Rothman ML, Williams GR, Lipscomb J, Matchar D. Determining clinically important differences in health status measures: a general approach with illustration to the Health Utilities Index Mark II. Pharmacoeconomics. 1999 Feb;15(2):141-55. https://doi.org/10.2165/00019053-199915020-00003 PMID: 10351188

2. Heyworth IT, Hazell ML, Linehan MF, Frank TL. How do common chronic conditions affect health-related quality of life?. British Journal of General Practice. 2009 Nov; 1;59(568):e353-8.

7. I am concerned about the conclusion “This study highlights the need for interventions or alternative treatment plans to minimize the use of multiple medications while improving the HRQoL of CKD patients”. Even though HRQoL is lower in heavier polypharmacy groups, how can we eliminate the possible claim that polypharmacy must have helped to maintain HRQoL at this level, and it is likely to become even lower if medication use is minimized in this multi-morbid population?

o Authors response: We appreciate the reviewer’s thoughtful comment. We agree that the association between polypharmacy and the HRQoL of CKD patients can be both complex and multifaceted. Furthermore, we understand that certain medications taken by patients with CKD can be used to improve the quality of life of these patients, however, polypharmacy becomes problematic when the reason for taking these medications is unclear. This becomes evident when medications are taken to address symptoms caused by other medications, leading to multiple adverse effects and potential drug interactions. 

o For instance, studies have shown that the use of 5 to 9 medications increases the chance of adverse reaction by 50%. This further increases by 100% when the number of medications is 20 or more [1]. In our current study 78% of the patients with CKD have either major polypharmacy (≥ 5 classes of medications) or excessive polypharmacy (≥ 9 classes of medication) and thus stand the risk for adverse drug reactions. 

o However, we have modified our statement in the abstract to read the following way (Line 51 to 51) 

 Reference

 1. Doan J, Zakrzewski-Jakubiak H, Roy J, Turgeon J, Tannenbaum C. Prevalence and risk of potential cytochrome P450-mediated drug-drug interactions in older hospitalized patients with polypharmacy. Ann Pharmacother. 2013;47(3):324-332. doi:10.1345/aph.1R621

8. Participants were selected from a large dataset that was generated using a self-administered questionnaire; ICD codes were used to identify those with CKD from that dataset. Have people self-reported their ICD codes in the questionnaire? If not from where those codes came?

o Authors response: Thank you for pointing this out. We have added more information to the data source section of the manuscript (Line number 125 to 136). Information identifying medical conditions were written as text and transcribed by professional medical coders following the format of the international classification of diseases, ninth and tenth revision, and clinical modification (ICD-9-CM and ICD-10-CM) codes. 

9. Excluded people with negative weight, negative PCS, negative MPS. How can they be negative (assume PCS and MPS were not re-scalled to Z scores)? Are you simply referring to the data cleaning process used to exclude invalid data entry in self-report? Also excluded those deceased. If the study is cross sectional (ie, data collected only once) you cannot have people in self-administrated data to exclude.

o Authors response: We acknowledge the concern regarding the omission of details explaining the exclusion of negative values and deceased patients from our analysis. Negative weight is a functionality in the process of cleaning the data. We have taken steps to address this by providing more information in the exclusion criteria subsection (Line number 143 to 153). 

10. Authors have checked for multicollinearity. Surprising that they didn’t find a presence of multicollinearity between polypharmacy and comorbidity count in that process, and also between comorbidity count and individual comorbidities.

o Authors response: We appreciate this comment. We did observe moderate pairwise correlation (estimate between 0.4 to < 0.6) between polypharmacy and comorbidity count and between comorbidity count and individual comorbidities. However, these correlations were not strong for the variables to be classified as being collinear to each other (since collinearity may likely exist if the parameter estimate for the correlation analysis is nearly 1 or -1 e.g greater than 0.8 or -0.8) [1]. 

o We did perform multicollinearity using variance inflation factor (VIF) and tolerance interval (TI) (Table 1) all explanatory values were within acceptable ranges [2-3].

References:

1. Kleinbaum DG, Kupper LL, Nizam A, Rosenberg ES. Applied regression analysis and other multivariable methods. Cengage Learning; 2013 Aug 30. (pg 

2. Daoud JI. Multicollinearity and Regression Analysis. JPhCS. 2017 Jan 2;949(1):012009.

3. Kim JH. Multicollinearity and misleading statistical results. Korean J Anesthesiol. 2019 Dec 1;72(6):558–69. https://doi.org/10.4097/kja.19087 PMID: 31304696

 

Table 1. Multicollinearity assessment (tolerance interval and variance inflation factor) of all explanatory variables. 

Variable Parameter Standard Error t Value Pr > |t| Tolerance Variance

Polypharmacy -0.41375 0.5611 -0.74 0.4612 0.7012 1.42614

Age Group 1.43996 0.53872 2.67 0.0077 0.86298 1.15877

Race/Ethnicity -0.23957 0.38856 -0.62 0.5378 0.88371 1.1316

Marital Status 0.06588 0.73532 0.09 0.9286 0.85989 1.16294

Income Level 1.45017 0.50028 2.9 0.0039 0.68752 1.45449

Prescription Drug Coverage -1.97478 1.16671 -1.69 0.091 0.3777 2.64759

Health Insurance Coverage 0.87983 1.03531 0.85 0.3958 0.37799 2.64558

Physical Activity -0.97537 0.79744 -1.22 0.2218 0.88136 1.13461

Access to Care -0.20149 1.34017 -0.15 0.8805 0.92621 1.07967

Census Region -0.35891 0.35786 -1 0.3163 0.94872 1.05405

Education 0.02267 0.77329 0.03 0.9766 0.85371 1.17135

Number of Comorbidities -0.4763 0.82737 -0.58 0.565 0.20815 4.80412

Depression -10.3288 1.04923 -9.84 <.0001 0.65687 1.52237

Severe Mental Disability -9.28216 1.25503 -7.4 <.0001 0.67455 1.48246

Anxiety 2.88347 0.96728 2.98 0.003 0.88176 1.13409

Diabetes 0.04958 0.95627 0.05 0.9587 0.50299 1.98811

Hypertension -0.02167 1.26456 -0.02 0.9863 0.61014 1.63897

Arthritis 0.06912 0.91035 0.08 0.9395 0.56347 1.77471

CVD -0.36581 0.9243 -0.4 0.6924 0.55044 1.81672

11. Should not describe non finding a statistically significant difference between groups as showing an equality (eg, “sex was equivalent to 50% of men and women”).

o Authors response: Thank you, we have corrected the manuscript to read as follows, “The mean age and standard deviation of CKD patients was 61.55 ± 13.93. Sex was not significantly different across the levels of polypharmacy” (Line number 233 to 234 of the manuscript). 

12. Figure 2. Arrange the legend in increasing order. Title refers to top 5 med classes but 8 in the figure. 

o Authors response: Thank you for identifying this. The legend in Figure 2 is now arranged in increasing order and the top 5 medications are listed. We have also included a statement that “the gray cells for each row or column should not be counted” since they are not applicable.

13. Authors have listed most commonly used drugs in some drug classes. Can this be explained by access issue (eg; cost, covered by insurance or public funds) rather than due to medical issues.

o Authors response: Thank you for this comment. We have controlled for health insurance and prescription drug coverage in our analysis. We have looked at the classes of medication by age group, low and high PCS and MCS values, polypharmacy categories and by sex (Fig 2). We believe the assessment of cost will be outside the scope of our current study. However, we have provided a brief assessment of the medications that are covered by insurance (See figure in Supplementary). 

14. Table 3. Please indicate reference category as such, instead of using “-“.

o Authors response: Replaced with a “Reference” in table 3.

15. Authors say “…, the PCS was four times lower among CKD patients with excessive polypharmacy than those with minor polypharmacy (ꞵ = -4.56, SE =0.28, p-value <0.0001)”. This is an incorrect interpretation of OLS regression coefficient

o Authors response: We have modified the manuscript to read as 4 points in place of 4 times (Line number 284 – 286, 342 – 343 and 314).

16. Also say “The results showed that polypharmacy may significantly lower the quality of life of non-dialysis CKD patients”. Please revise this to eliminate implied causation, study haven’t assessed the causation.

o Authors response: This has been modified. All statements with implied causation have been modified to indicate association. 

17. Table 2 footnote: what is health polypharmacy (is the sentence incomplete?).

o Authors response: This grammatical oversight has been corrected in Table 2. 

18. When presenting something in X ±Y format, please be specific about what is X and what is Y (eg; “mean age was 61.55±13.93”; here 13.93 is SD or SE or IQR, or something else?).

o Authors response: We have clarified the manuscript in the results section to read,“The mean age and standard deviation of CKD patients was 61.55 ± 13.93.”

19. Table 1 & 2 footnotes. Please remove notes that are not applicable to that table (eg, CI in table1, W in table2).

o Authors response: All notes that are not applicable to table 1 and 2 have been removed. 

20. Unrealistic level of precision (ie, too many DPs) in p- values in tables and text.

o Authors response: We have modified the DPs from 4 DPs as it was previously to 3 DPs. 

21. Replace multivariate with multivariable, this study haven’t used multivariate models.

o Authors response: The word multivariate has been has been replaced with multivariable. 

Continuation of Reviewers comment (Summary of Reviewer #1 comments) 

22. There are however several issues that should be addressed prior to consideration of publication: The fundamental issues with this paper is whether the polypharmacy is the cause or a consequence of the reduced HRQoL. The authors seem to claim it is the polypharmacy that reduces the HRQoL, but this is impossible to know from this study. There is purely an association which is not surprising.

o Authors response: We acknowledge your comment on this, and we do agree. We have now clearly delineated in the limitation section that polypharmacy does not lead to reduced HRQoL since this is not a cause-and-effect study. We have modified all wordings and statements that tend to imply causation with wordings and statements that depict an association study.

23. The patient selection and timing of the surveys and data collection must be more clearly stated. Were the medication data and HRQoL surveys done simultaneously? 

o Authors response: The medication data and health data were done simultaneously with some slight differences. Data across all years were consolidated to obtain total participants with CKD. All medications taken by each individual CKD patient were sorted using the prescription medicine file leading to a much larger data since most individuals in our study were prescribed multiple classes of medications taken concomitantly over a given round. 

o All statistical analysis was conducted using the yearly consolidated data files. But the analysis for the classes of medications were conducted using prescription medicine data file that accounts for all medications taken by each patient in our study. 

24. Why were there so few patients with CKD enrolled (0.2% - when the prevalence is 15% in the population?) Were these all-stage IV CKD?

o Authors response: This is a very important observation, and we acknowledge the reviewer for this comment. A possible reason for the low number of participants with CKD could be due to the self-reported nature of the MEPS data. Since laboratory results such as glomerular filtration rate (GFR) was not obtained from participants, it is possible that some participants with CKD may not know they have CKD. 

o For instance, according to the center for disease control (CDC) [1], 9 out of 10 individuals are unaware of their CKD status. Since only 10% are aware of their CKD condition then 10% of 14% (prevalence rate of CKD in the United States) = 1.4% which is much closer to the estimate we observed from the MEPS data of 0.27% and have been explained in the manuscript (Line number 352 to 358).

Reference:

Centers for Disease Control and Prevention. Chronic Kidney Disease in the United States. Atlanta, GA: US Department of Health and Human Services, Centers for Disease Control and Prevention. 2023 [cited 2023 Aug 22]. Available from: https://www.cdc.gov/kidneydisease/publications-resources/ckd-national-facts.html

25. Why were only DM, HT and arthritis considered as comorbidities? Most CKD patients with die of cardiovascular disease before needing dialysis, this would have been an important group to report on.

o Authors response: Thank you for pointing this out. We have now adjusted for cardiovascular disease (CDV) in the model. 

26. What was the definition of “access to care” used? 

o Authors response: In the manuscript we have added the following as a description to access to care (Line number 197 to 199). “Access to care was used to determine if a patient had a designated location, such as a doctor's office or health care facility, where they can go to receive care if they are ill or require guidance regarding their health condition.” 

27. Whether the indeed number of medications “excessive” or not is also a question. Patients with CKD do have many medications, in many cases because of the associated comorbidities, but also with increasing severity of kidney disease, as one tries to stave off the need for dialysis. I am in full agreement that polypharmacy should be examined in each patient and the medications rationalized as much as possible, sometimes it is not possible. Another word should be chosen rather than “excessive”. Avoid loaded phrases like “patients being constantly prescribed…” etc…

o Authors response: We thank the reviewer for pointing this out. The phrase has been modified (Line number 321 to 323). Additionally, referring to the word “excessive”, in a systematic review on polypharmacy [1], 7 out of 10 articles that examined individuals taking 10 or more unique therapeutic classes of medications used the term “excessive polypharmacy,” while 2 others used the term “Severe Polypharmacy”, and one other used the term “Hyperpolypharmcy.” For the current study, we decided to use the term “excessive polypharmacy” since it is the most commonly used term in the literature. 

o Reference:

1. Masnoon N, Shakib S, Kalisch-Ellett L, Caughey GE. What is polypharmacy? A systematic review of definitions. BMC Geriatr. 2017 Oct 10;17(1):1–10. https://doi.org/10.1186/s12877-017-0621-2 PMID: 29017448

28. Was there any way to assess stage of CKD and or number of comorbidities here, as these would be major confounders of assessment of HRQoL? The dialysis population may be more uniform in this sense. 

o Authors response: We appreciate this comment. Unfortunately, due to the limitations of the MEPS dataset, we were unable to assess the stages of CKD but we were able to evaluate the number of comorbidities (Table 1-3). 

29. Was it possible to derive the pill burden in this study? Some medications e.g. sodium bicarbonate or phosphate binders may be two class of medication but may bring 9 tablets a day each. The numbers of pills may well reduce HRQoL and some classes more than others (e.g. phosphate binders). It would have been of interest to assess I there is a relationship between drug the classes and HRQoL. Pill burden and number of medication classes cannot be used interchangeably.

o Authors response: Unfortunately, we were unable to assess pill burden due to limitation of the MEPS data. We do agree this would have been extremely useful to examine. However, we plan on assessing the drug classes and HRQoL in another study and we do appreciate this recommendation. 

30. In the tables it would make more sense to have the percentages across the horizontal than the vertical, to understand what proportion of e.g. male patients had which degree of polypharmacy (I.e. add up to 100 horizontally and not vertically) 

o Authors response: We agree and have updated table 1 to read up to hundred percent horizontally. 

 

Reviewer #2

1. The biggest limitation of this study is its observational nature that make drawing conclusions about causation difficult, even allowing for the use of an adjusted multifactorial model. It remains likely that the higher use of the medications in patients with lower QOL may be largely explained by the fact that patients with a lower QOL have more medical problems and therefore need more medication. I don’t believe that we can easily conclude from these data that if patients with multiple health problems were prescribed fewer medications that their QOL would necessarily improve. To determine that would require a prospective and interventional study. I believe this limitation should be acknowledged in the discussion.

o Authors response: We wholeheartedly agree with the reviewer on this comment and for this reason we have modified our study correcting wordings and statements that imply causation with statements that imply association. Additionally, in the limitation section we have clearly delineated that this finding does not imply causation (Line number 383 to 385). 

2. There should be some acknowledgement in the discussion that polypharmacy is not the only potentially inappropriate prescribing practice that CKD patients are at risk of. For example, Smits et al (BMJ Open, July 2019) described that both overprescribing and underprescribing were problems in CKD patients. For example, they identified underuse of RAAS inhibitors and statins in some patients who would be predicted to benefit from these medications. Another form of inappropriate prescribing in inappropriate adjustment of medication dosage in patients with CKD, as identified by Castelino et al. (BMC Nephrology, June 2020), however, the methodology in the present study by Adjeroh was unable to address this aspect, as creatinine clearance and eGFR were not assessed. 

o Authors response: We do agree with the sentiments in this inquiry. Unfortunately, we were unable to assess eGFR. We adjusted the statement in the manuscript (Line number 327 to 332). 

 

3. As per Fig 1, the rate of self-reported CKD in the patient population is much lower than would be predicted (i.e. 932/339,883 = 0.27%), considering that approximately 10% of the adult population have CKD. This observation raises questions of selection and recall bias and may influence the generalisability of the conclusions. This need to be acknowledged and discussed in the manuscript.

o Authors response: (Similar response from comment #24 above). This is a very important observation, and we acknowledge the reviewer for this comment. A possible reason for the low number of participants with CKD could be due to the self-reported nature of the MEPS data. Since laboratory results such as glomerular filtration rate (GFR) was not obtained from participants, it is possible that some participants with CKD may not know they have CKD. 

o For instance, according to the center for disease control (CDC) [1], 9 out of 10 patients are unaware of their CKD status. Since only 10% are aware of their CKD condition then 10% of 14% (prevalence rate of CKD in the United States, 2023) = 1.4% which is much closer to the estimate we observed from the MEPS data of 0.27% and have been explained in the manuscript (Line number 352 to 358).

Reference:

Centers for Disease Control and Prevention. Chronic Kidney Disease in the United States. Atlanta, GA: US Department of Health and Human Services, Centers for Disease Control and Prevention. 2023 [cited 2023 Aug 22]. Available from: https://www.cdc.gov/kidneydisease/publications-resources/ckd-national-facts.html

4. In figure 3 (top 10 medications) a mixture of generic and brand names for the medications is used. This should be corrected to only use the generic drug names. Moreover, in several cases the same drug is listed twice in a graph with the generic and brand names being listed as different drugs (e.g. 3D atorvastatin and Lipitor, rosuvastatin and Crestor, 3E furosemide and Lasix). Please address this.

o Authors response: We thank the reviewer for identifying this transcript error. We have thoroughly gone through the entire prescription file with expert guidance, the use of the National ambulatory medical care survey and national hospital medical care survey database that provides information about all prescription and nonprescription drug products in the united state. We used these mediums to ensure that all generic prescription medications are coded correctly, and we have modified all figures with correct classes of medications and prescription drug information. 

Minor comments from reviewer #2.

5. Line 105 (Research Design) suggest “associations” rather than “effects”.

o Authors response: This has been corrected.

6. I have difficulty understanding this sentence (Data Analysis, line 184-186) “The missing data from the three explanatory variables (education, access to care, and physical activity) were disregarded in the model since no discernible missing pattern was observed (this was assessed using proc mi in SAS), with the overall missingness of these variables accounting for less than 5% of the eligible participants”. Please review and consider revising this sentence.

o Authors response: We apologize for the confusion. However, we have modified the statement. All we wanted to say is missing data was addressed using pairwise deletion and has been modified in the manuscript (Line number 226). We used pairwise deletion because we observed the missing data pattern using proc mi in SAS and it showed that the data was missing completely at random. We also applied Little’s missing completely at random test (p-value = 0.602 > alpha of 0.05) which equally confirmed data was missing completely at random. Since the number of missing values were less than 5% and there were no missing values in the dependent and independent variables, pairwise deletion was applied.

Reference

Little, R. J. A. 1988. A test of missing completely at random for multivariate data with missing values. Journal of the American Statistical Association 83: 1198–1202.

7. Line 244, pg 16. I think this should be “four points” instead of “four times”. Please review.

o Authors response: This has now been corrected (Line number 284 – 286, 342 – 343 and 314). 

8. Line 277, pg 19. Suggest change “the effects of polypharmacy on the HRQoL”, to “the associations between polypharmacy and HRQoL”.

o Authors response: This has been corrected.

9. 2 Line 303, pg 20. Suggest change “plausible” to relevant”.

o Authors response: This has been corrected.

10. Line 314, pg 21. Hydrocodone is also an opioid.

o Authors response: We do agree.

11. Line 315-317. Please review this sentence “Although, it was shown that analgesics such as opioids were used in advanced stages of CKD for patients 65 years and older [36], which contrasts with our study findings”. I am not clear what you are trying to say here.

o Authors response: We have revised the statement (Line number 368 to 372)

12. Line 322. Recommend change “increased the risk” to “was associated with an increased risk”.

o Authors response: It has been modified based on the reviewer’s suggestion.

---

## [Decision Letter · Decision Letter 1]

2 Oct 2023

PONE-D-23-16559R1The association between polypharmacy and health-related quality of life among non-dialysis chronic kidney disease patientsPLOS ONE

Dear Dr. Adjeroh,

Thank you for submitting your manuscript to PLOS ONE. After careful consideration, we feel that it has merit but does not fully meet PLOS ONE’s publication criteria as it currently stands. Therefore, we invite you to submit a revised version of the manuscript that addresses the points raised during the review process.

We look forward to receiving your revised manuscript.

Kind regards,

Ari Samaranayaka, PhD

Academic Editor

PLOS ONE

Journal Requirements:

Additional Editor Comments:

Authors have taken a large effort to response to my concerns, most responses are satisfactory. However, following few places are still unclear. I assume rephrasing relevant sentences can improve the clarity to readers.

(1) In last review I questioned about the polypharmacy groupings. Authors say, “we classified polypharmacy based on the number of unique therapeutic classes of prescribed medications taken concomitantly over a specific round (e.g., 6-month data collection period) for each year”. According to study description, same person can be captured in data in multiple years. For a given person, the number of medications likely to fluctuate over time, therefore the person will be grouped into different polypharmacy groups in different years. This creates major implications on statistical analysis because chisquare test and OLS regression etc require each person be in a single polypharmacy group. How was this handled in the statistical analyses?

(2) In the last review I asked if the list of most commonly used drugs can be driven by the access issues (eg; cost, covered by insurance or public funds, etc) rather than by the medical issues. As a response authors say, “we have controlled for health insurance and prescription drug coverage in our analysis”. I am not willing to accept this explanation because statistical controlling is relevant when presenting something estimated from a (multivariable) statistical model, it is irrelevant when presenting a descriptive list as observed in data. However, I like to accept their next explanation that says “We believe the assessment of cost will be outside the scope of our current study”.

Reviewers' comments:

Reviewer's Responses to Questions

**Comments to the Author**

1. If the authors have adequately addressed your comments raised in a previous round of review and you feel that this manuscript is now acceptable for publication, you may indicate that here to bypass the “Comments to the Author” section, enter your conflict of interest statement in the “Confidential to Editor” section, and submit your "Accept" recommendation.

Reviewer #1: (No Response)

Reviewer #2: (No Response)

2. Is the manuscript technically sound, and do the data support the conclusions?

Reviewer #1: Yes

Reviewer #2: Yes

3. Has the statistical analysis been performed appropriately and rigorously? 

Reviewer #1: I Don't Know

Reviewer #2: Yes

4. Have the authors made all data underlying the findings in their manuscript fully available?

Reviewer #1: No

Reviewer #2: Yes

5. Is the manuscript presented in an intelligible fashion and written in standard English?

Reviewer #1: Yes

Reviewer #2: Yes

6. Review Comments to the Author

Reviewer #1: The authors have taken the comments on board and have largely improved the manuscript. The limitations are now clearer. I still struggle with the definition "excessive" polypharmacy, as the indications for the various medications were not known.

Also I would consider a major conclusion to be that the pill burden is high among patients with a known diagnosis of CKD, and that given this it is possible that some of this polypharmacy may be "excessive", therefore clinicians should be aware and review the medication list regularly. Also a prospective study is required to assess the associated pill burden and deeper investigation into the contributors ot reduced quality of life may help to understand the good or bad effect of the prescriptions.

Reviewer #2: I am satisfied that nearly all of the concerns I raised have been addressed.

The only outstanding issue to correct is that I see an error in the revision is response to my second point about medication under-prescribing (lines 327-332). This should refer to UNDER prescribing of RAAS inhibitors, Vit D and statins, and OVER prescribing of NSAIDs. At present, this appears the wrong way around.

7. PLOS authors have the option to publish the peer review history of their article (what does this mean?). If published, this will include your full peer review and any attached files.

Reviewer #1: No

Reviewer #2: No

---

## [Author Response · Author response to Decision Letter 1]

18 Oct 2023

We extend our gratitude for the second time to the reviewers for generously dedicating their valuable time and expertise towards reviewing our manuscript. Their insightful review has made a significant contribution to improving the quality of this manuscript. We have combined all comments again and meticulously addressed each comment in accordance with the reviewers’ recommendations. The remarks from the reviewers have been formatted in italics for clarity. The changes can be seen in track changes in the file “Revised Manuscript with Track Changes.”

Reviewer #1

1. In last review I questioned about the polypharmacy groupings. Authors say, “we classified polypharmacy based on the number of unique therapeutic classes of prescribed medications taken concomitantly over a specific round (e.g., 6-month data collection period) for each year”. According to study description, same person can be captured in data in multiple years. For a given person, the number of medications likely to fluctuate over time, therefore the person will be grouped into different polypharmacy groups in different years. This creates major implications on statistical analysis because chisquare test and OLS regression etc require each person be in a single polypharmacy group. How was this handled in the statistical analyses?.

o Authors response: Thank you for the opportunity to clarify this point. Each year the Agency for Healthcare Research and Quality (AHRQ) releases two sets of data: The consolidated data and a longitudinal data file. The consolidated yearly released data files are designed such that individuals that appeared in the current year data are independent from individuals that appear in the previous year data. Each participant is assigned unique weights and variance estimation strata. Excluding individuals from the data with the assumption that they appeared in the previous year will completely compromise the integrity of the data. 

o On the other hand, the released longitudinal datasets are such that an individuals can appear in two consecutive years. With such a scenario, AHRQ assigns similar weights and variance estimation strata indicating that persons appearing in the previous year and current year are similar. 

o For this study, we utilized the consolidated yearly released data files indicating that patients who appeared in the previous year's data files are independent of patients who appeared in the current year. We couldn't use the longitudinal data file because the number of individuals in the follow-up year was very small, leading to a much smaller sample size (Line number 118 to 121 and 170 to 172).

2. In the last review I asked if the list of most commonly used drugs can be driven by the access issues (eg; cost, covered by insurance or public funds, etc) rather than by the medical issues. As a response authors say, “we have controlled for health insurance and prescription drug coverage in our analysis”. I am not willing to accept this explanation because statistical controlling is relevant when presenting something estimated from a (multivariable) statistical model, it is irrelevant when presenting a descriptive list as observed in data. However, I like to accept their next explanation that says “We believe the assessment of cost will be outside the scope of our current study”.

o Authors response: Thank you for calling our attention to this important additional information. As a supporting document in the manuscript, we have included a chart on insurance coverage by drug (S4 Fig). Also see chart below.

S4 Fig. Top 5 Medications used by non-dialysis CKD patients by therapeutic drug class with and without insurance coverage (percentage (%) = weighted percentage).

3. The authors have taken the comments on board and have largely improved the manuscript. The limitations are now clearer. I still struggle with the definition of "excessive" polypharmacy, as the indications for the various medications were not known.

o Authors response: Thank you for pointing this out again. Another term that is also used in the literature is hyperpolypharmacy (see the listed articles below). All terms with excessive polypharmacy in the manuscript have been changed to hyperpolypharmacy. 

o References:

Masnoon N, Shakib S, Kalisch-Ellett L, Caughey GE. What is polypharmacy? A systematic review of definitions. BMC geriatrics. 2017 Dec;17:1-0.

Palmer K, Villani ER, Vetrano DL, Cherubini A, Cruz-Jentoft AJ, Curtin D, Denkinger M, Gutierrez-Valencia M, Guðmundsson A, Knol W, Mak DV. Association of polypharmacy and hyperpolypharmacy with frailty states: a systematic review and meta-analysis. European Geriatric Medicine. 2019 Feb 15;10:9-36.

Herrinton LJ, Lo K, Alavi M, Alexeeff SE, Butler KM, Chang C, Chang CC, Chu VL, Krishnaswami A, Deguzman LH, Prausnitz S. Effectiveness of Bundled Hyperpolypharmacy Deprescribing Compared With Usual Care Among Older Adults: A Randomized Clinical Trial. JAMA Network Open. 2023 Jul 3;6(7):e2322505-..

Kennel PJ, Kneifati-Hayek J, Bryan J, Mehta K, Banerjee S, Sobol I, Safford M, Goyal P. Prevalence and Determinants of Hyperpolypharmacy in Adults with Heart Failure. Journal of Cardiac Failure. 2018 Aug 1;24(8):S33.

 

4. Also, I would consider a major conclusion to be that the pill burden is high among patients with a known diagnosis of CKD, and that given this it is possible that some of this polypharmacy may be "excessive", therefore clinicians should be aware and review the medication list regularly. Also a prospective study is required to assess the associated pill burden and deeper investigation into the contributors ot reduced quality of life may help to understand the good or bad effect of the prescriptions.

o Authors response: Thank you for this additional useful statement. It has been included in the manuscript in line number 402 to 406. 

Reviewer #2

1. The only outstanding issue to correct is that I see an error in the revision is response to my second point about medication under-prescribing (lines 327-332). This should refer to UNDER prescribing of RAAS inhibitors, Vit D and statins, and OVER prescribing of NSAIDs. At present, this appears the wrong way around.

o Authors response: Thank you for pointing out this transcript error. We have corrected it in the manuscript (Line number 329 to 332). 

We have included all datasets that we worked with at this point (analytical and medication dataset). We do not have any other dataset at our disposal. All datasets that we have worked with can be publicly assessed from the medical panel survey data website. https://www.ahrq.gov/data/meps.html

In the manuscript, the link to reference list #2 was changed from https://dph.georgia.gov/sites/dph.georgia.gov/files/related_files/site_page/kidney_factsheet%20(1)%20cdc.pdf to https://dpacmi.org/documents/kidney_Factsheet.pdf (The previous link can no longer be found). The same information was found in the new link.

---

## [Decision Letter · Decision Letter 2]

23 Oct 2023

The association between polypharmacy and health-related quality of life among non-dialysis chronic kidney disease patients

PONE-D-23-16559R2

Dear Dr. Adjeroh,

We’re pleased to inform you that your manuscript has been judged scientifically suitable for publication and will be formally accepted for publication once it meets all outstanding technical requirements.

Kind regards,

Ari Samaranayaka, PhD

Academic Editor

PLOS ONE

Additional Editor Comments (optional):

Reviewers' comments:

Reviewer's Responses to Questions

**Comments to the Author**

1. If the authors have adequately addressed your comments raised in a previous round of review and you feel that this manuscript is now acceptable for publication, you may indicate that here to bypass the “Comments to the Author” section, enter your conflict of interest statement in the “Confidential to Editor” section, and submit your "Accept" recommendation.

Reviewer #1: All comments have been addressed

Reviewer #2: All comments have been addressed

2. Is the manuscript technically sound, and do the data support the conclusions?

Reviewer #1: Yes

Reviewer #2: Yes

3. Has the statistical analysis been performed appropriately and rigorously? 

Reviewer #1: I Don't Know

Reviewer #2: Yes

4. Have the authors made all data underlying the findings in their manuscript fully available?

Reviewer #1: Yes

Reviewer #2: Yes

5. Is the manuscript presented in an intelligible fashion and written in standard English?

Reviewer #1: Yes

Reviewer #2: Yes

6. Review Comments to the Author

Reviewer #1: no further comments. The manuscript has been improved. The authors have taken the comments on board.

Reviewer #2: My previous comments about under and over prescribing have now all been addressed by the authors. I have no new concerns to raise.

7. PLOS authors have the option to publish the peer review history of their article (what does this mean?). If published, this will include your full peer review and any attached files.

Reviewer #1: No

Reviewer #2: No

---

## [Editor Report · Acceptance letter]

3 Nov 2023

PONE-D-23-16559R2 

The association between polypharmacy and health-related quality of life among non-dialysis chronic kidney disease patients 

Dear Dr. Adjeroh:

I'm pleased to inform you that your manuscript has been deemed suitable for publication in PLOS ONE. Congratulations! Your manuscript is now with our production department. 

Kind regards, 

on behalf of

Dr. Ari Samaranayaka 

Academic Editor

PLOS ONE